# Microbiome and Long COVID-19: Current Evidence and Insights

**DOI:** 10.3390/ijms262010120

**Published:** 2025-10-17

**Authors:** Olga A. Caliman-Sturdza, Sevag Hamamah, Oana C. Iatcu, Andrei Lobiuc, Anca Bosancu, Mihai Covasa

**Affiliations:** 1College of Medicine and Biological Science, Stefan cel Mare University of Suceava, 720229 Suceava, Romania; olga.caliman-sturdza@usm.ro (O.A.C.-S.); oana.iatcu@usm.ro (O.C.I.); andrei.lobiuc@usm.ro (A.L.); anca.bosancu@usm.ro (A.B.); 2Emergency Clinical Hospital Suceava, 720224 Suceava, Romania; 3Department of Internal Medicine, Scripps Mercy Hospital, San Diego, CA 92103, USA; hamamah.sevag@scrippshealth.org

**Keywords:** SARS-CoV-2, COVID-19, gut microbiota, dysbiosis, probiotics, SCFA

## Abstract

Long COVID, also referred to as post-acute sequelae of SARS-CoV-2 infection (PASC), is characterized by persistent multi-systemic symptoms such as fatigue, cognitive impairment, and respiratory dysfunction. Accumulating evidence indicates that gut and oral microbiota play an important role in its pathogenesis. Patients with long COVID consistently exhibit reduced microbial diversity, depletion of beneficial short-chain fatty acid (SCFA)-producing species such as *Faecalibacterium prausnitzii* and *Bifidobacterium* spp. and enrichment of proinflammatory taxa including *Ruminococcus gnavus*, *Bacteroides vulgatus*, and *Veillonella*. These alterations may disrupt intestinal barrier integrity, sustain low-grade systemic inflammation, and influence host immune and neuroendocrine pathways through the gut–brain and gut–lung axes. Distinct microbial signatures have also been associated with symptom clusters, including neuropsychiatric, respiratory, and gastrointestinal manifestations. Proposed mechanisms linking dysbiosis to long COVID include impaired SCFA metabolism, tryptophan depletion, microbial translocation, and interactions with host immune and inflammatory responses, including autoantibody formation and viral antigen persistence. Preliminary interventional studies using probiotics, synbiotics, and fecal microbiota transplantation suggest that microbiome-targeted therapies may alleviate symptoms, although evidence remains limited and heterogeneous. This review synthesizes current literature on the role of gut and oral microbiota in long COVID, highlights emerging microbial biomarkers, and discusses therapeutic implications. While causality remains to be firmly established, restoring microbial balance represents a promising avenue for diagnosis, prevention, and management of long COVID.

## 1. Introduction

Long COVID-19, also referred to as post-acute sequelae of COVID-19 (PASC), is increasingly recognized as a major public health challenge. It encompasses a constellation of persistent symptoms such as fatigue, cognitive impairment (“brain fog”), shortness of breath, and sleep disturbances that may last for weeks or months following acute SARS-CoV-2 infection [1]. Although awareness of the condition has grown, its pathogenic underpinnings remain incompletely defined. Current hypotheses implicate viral persistence, immune dysregulation, autoantibody production, endothelial and microvascular injury, mitochondrial dysfunction, and reactivation of latent viruses [2,3]. Emerging evidence also highlights the microbiome as a critical, yet underexplored, factor in long COVID. The gut and oral microbiota are central modulators of immune and inflammatory processes [4,5], and disturbances in microbial balance (dysbiosis) during acute infection may persist and are associated with chronic low-grade inflammation and prolonged symptomatology [6,7]. Several recent studies have reported alterations in microbial diversity, enrichment of pathogenic taxa, and depletion of beneficial commensals in individuals with long COVID, suggesting that the microbiome may serve as potential biomarker of disease and could be linked to symptom persistence. This review goes beyond prior long COVID microbiome studies by integrating the gut–oral–lung axes to provide a systemic perspective on microbial dysbiosis. It consolidates evidence on specific microbial taxa, functional genes, and short-chain fatty acid (SCFA)-related metabolic pathways as candidate biomarkers, highlighting their potential for diagnostic and prognostic use. Additionally, it explores translational applications, including dietary interventions, probiotics, synbiotics, and fecal microbiota transplantation, emphasizing how microbiome-targeted strategies could modulate persistent symptoms and guide personalized therapy. By connecting mechanistic insights with clinical interventions, this review offers a comprehensive framework that bridges basic research and potential clinical practice.

## 2. Gut Microbiome Alterations in Various Phases of Long COVID

### 2.1. Altered Gut Microbiome During COVID-19 Infection

COVID-19 has been shown to cause changes to the human gut microbiome, both in the short and long terms, potentially contributing to its pathogenesis. During the acute phase, beneficial bacterial genera species including *Faecalibacterium prausnitzii*, *Eubacterium rectale* and *Bifidobacterium* are significantly underrepresented during COVID-19 infections and up to 30 days thereafter. These changes have been associated with elevated concentrations in inflammatory markers that correlate with disease severity [8]. Similarly, during time of hospitalization, patients exhibited significantly altered microbiomes, with notable decline of beneficial bacteria and enrichment of unfavorable pathobionts [9]. *Faecalibacterium* again displayed inverse correlation with disease severity, with relative depletion of this anti-inflammatory genus when compared to controls. Conversely, there was an overabundance of opportunistic pathogenic microbiota such as *Coprococcus*, *Clostridium ramosum*, and *Clostridium hathewayi*, which correlated with disease severity during hospitalization [9]. These changes also persisted in the short-term despite resolution of acute phase symptoms. Further, study findings have shown that individuals that develop long COVID harbor distinct gut microbial composition during acute infection, that differed from both individuals that were COVID-19-positive with similar symptoms who did not develop long COVID and healthy controls [10]. Patients that developed long COVID displayed increased abundances of certain microbes from families Anaerovoracaceae and Acutalibacteraceae, as well as genera *Blautia*, *Schaedlerella*, *Massilimaliae*, *Faecalimonas*, and *Ruthenibacterium*. Microbial taxa such as *Eubacterium*, *Lachnoclostridium*, *Prevotella*, *Carnobacterium*, and *Weissella* and the family Lachnospiraceae were associated with subsequent long COVID development, and in this study they appeared more strongly linked to outcomes than other biomarkers such as fecal calprotectin.

### 2.2. Lasting Changes in Microbial Disruption in Long COVID

Beyond the acute phase, there is growing evidence that COVID-19 infection is associated with long-lasting alterations in gut microbial composition, an effect that was shown in studies ranging from three months up to one year after initial infection [11,12,13]. For example, in a three-month follow-up study where microbial diversity was assessed in post-acute COVID-19 patients, there was a decrease in the Chao index that estimate species richness, Simpson index (a measure of diversity) and α-diversity, which collectively indicates unevenness and microbial depletion [14]. Six-month follow-up has also shown persistently reduced gut microbial richness and distinct microbial compositions compared to uninfected controls [15]. Even mild COVID-19 infection has been associated with significantly deplete *Akkermansia muciniphila*, an important mucin-degrading bacterium that maintains gut barrier integrity. Recent findings have shed further light on the prevalence of two distinct enterotypes identified during the acute phase of COVID-19 and tracked for up to two years post-recovery [16]. These include a *Blautia*-dominated enterotype (Enterotype-B) and a *Streptococcus*-dominated enterotype (Enterotype-S). Patients with Enterotype-B exhibited greater species richness and diversity, along with lower levels of inflammation and less severe acute-phase infection. In contrast, individuals with a persistent Enterotype-S at six months showed a higher incidence of residual pulmonary abnormalities on computed tomography (CT) scans, observed in 55% of Enterotype-S cases compared to 20% in Enterotype-B. While these studies did not specifically focus on long COVID patients, they offer valuable insight into the lasting microbial alterations observed in the post-acute phase of COVID-19.

In general, individuals with long COVID have decreased microbial diversity and a significant relative decline in concentrations of beneficial bacterial species, similar to what has been noted in the acute and post-acute phases [13]. Prospective studies have further clarified key microbial genera associated with long COVID during the six-month post-infection period [17]. In one study analyzing the fecal microbiota of 106 patients with persistent symptoms, there was a decrease in *F. prausnitzii* and *Bifidobacterium* spp., alongside an increase in *Ruminococcus gnavus* and *Bacteroides vulgatus*. Notably, the butyrate-producing species *F. prausnitzii* and *Bifidobacterium pseudocatenulatum* showed the strongest inverse correlation with the development of long COVID. More recently, additional studies have confirmed persistent microbial instability up to one year after hospital discharge, with dysbiosis closely associated with ongoing long COVID symptoms [13].

Across these studies, patients who fully recovered from COVID-19 without residual symptoms often showed normalization of their gut microbiota [13,17,18,19,20] (Figure 1). For instance, in one study conducted six months after acute infection, asymptomatic individuals who did not develop long COVID had a gut microbial composition comparable to that of healthy controls [17]. In contrast, individuals with persistent symptoms frequently exhibited gut microbiome alterations, including reduced diversity and shifts in the abundance of key bacterial taxa [6,15,21]. These findings suggest that while some individuals experience prolonged gut dysbiosis, others undergo microbiota normalization, potentially due to internal or external factors [22]. However, the specific mechanisms or risk factors that determine whether a patient will recover or continue to experience microbial instability after acute COVID-19 remain poorly understood.

### 2.3. Trends in Taxonomic Shifts in Gut Microbiota

Current literature reveals consistent taxonomic alterations in the gut microbiota of individuals with long COVID. Among the most frequently reported changes is a de-creased abundance of *F. prausnitzii*, that may begin during the acute phase of infection and persist over time [8,17,23]. Similarly, *Bifidobacterium* and *Eubacterium* spp. have been repeatedly shown to be reduced in long COVID patients compared to healthy controls [24,25,26]. Notably, higher levels of *Bifidobacterium* are inversely correlated with the development of long COVID [17]. Furthermore, in a large gut metagenomic study involving 2871 participants, depletion of *E. rectale* was associated with increased COVID-19 mortality [27]. These genera are well-documented producers of SCFAs, particularly butyrate, which exert anti-inflammatory effects, mechanisms discussed in more detail in a subsequent section. In contrast, long COVID has been associated with enrichment of proinflammatory taxa such as *R. gnavus*, *B. vulgatus*, and *Clostridium* spp. [9,17,28]. Collectively, these taxonomic shifts may promote increased expression of proinflammatory cytokines and disrupt intestinal barrier integrity, contributing to systemic inflammation and the persistence of long COVID symptoms.

The studies summarized in Table 1 provide evidence of persistent microbiome alterations in long COVID; however, many are limited by small sample sizes, observational designs, and lack of control for confounding factors, highlighting the need for cautious interpretation.

### 2.4. Oral and Respiratory Microbiomes

Emerging research suggests that microbiomes beyond the gut, particularly in the oral and respiratory tracts, may be associated with the development and persistence of long COVID. Factors such as poor oral hygiene, smoking, and alcohol use have been linked to alterations in the oral microbiome and may influence study findings [34]. At the phylum level, COVID-19 patients show increased abundances of Firmicutes and Proteobacteria, while healthy controls have higher levels of Bacteroidetes. At the genus level, *Streptococcus* and *Veillonella* are more prevalent in COVID-19 patients, whereas *Prevotella* is more abundant in non-COVID-19 controls, patterns that mirror the broader phylum-level shifts. Further evidence from a separate study comparing COVID-19 patients with and without long COVID revealed an enrichment of potentially pathogenic taxa, including Campylobacterota, Coriobacteriales, Pseudomonadales, and Campylobacterales [35]. Similarly to the gut microbiota, the oral and respiratory microbiomes in COVID-19 patients exhibit reduced microbial diversity, as indicated by lower Shannon diversity index scores and Chao1 richness estimates [34]. Moreover, longitudinal studies have demonstrated that oral dysbiosis can persist well beyond viral clearance, even in patients who received antibiotics during the acute phase of infection [36]. These findings underscore the potential role of sustained microbial imbalance in the oral and respiratory tracts in the pathophysiology of long COVID. Additionally, Haran and colleagues investigated outpatient COVID-19 cases in individuals vaccinated with mixed vaccine regimens. They found that 37% of these patients went on to develop long COVID and identified distinct microbial profiles in their oral cavities [33,37,38]. Specifically, patients who developed long COVID had elevated levels of certain *Prevotella* and *Veillonella* species, lipopolysaccharides (LPS)-producing bacteria known to stimulate proinflammatory cytokine production [33,39]. It is important to note, however, that these findings regarding *Prevotella* abundance contrast with those of a previous study [34], which reported higher *Prevotella* levels in non-COVID-19 controls. This discrepancy may be attributed to differences in study design and timing, as the earlier study focused on patients during the acute phase of infection, whereas Haran et al. examined microbial profiles during the post-acute phase. Nevertheless, these bacteria can be aspirated into the lower respiratory tract and have been associated with systemic inflammation [40,41,42,43]. The study led by Haran further suggested that the oral microbiomes of individuals with long COVID resemble those observed in patients with myalgic encephalomyelitis/chronic fatigue syndrome (ME/CFS). The same study also presented evidence that, in individuals with long COVID, bacterial products from the gut or lungs may translocate into the bloodstream, contributing to systemic effects [44]. Currently, research on the nasopharyngeal and pulmonary microbiomes in the context of long COVID remains limited. While it is well-established that acute COVID-19 affects the microbial communities of the upper airways [45], it is still unclear whether dysbiosis in the respiratory tract persists in long COVID and contributes to ongoing symptoms, such as prolonged anosmia or pulmonary dysfunction. Further studies are needed to clarify the potential role of respiratory microbiome alterations in the pathogenesis and symptomatology of long COVID. This topic is explored further in the following subsection.

### 2.5. Unique Microbial Clusters Associated with Long COVID Symptoms

Emerging evidence suggests that specific alterations in gut microbiota are linked to distinct long COVID symptom profiles. For example, in a six-month follow-up study from Hong Kong, neuropsychiatric symptoms and fatigue were associated with the enrichment of pathobionts such as *C. innocuum* and *Actinomyces naeslundii* [17]. Similarly, persistent respiratory symptoms were linked to increased levels of opportunistic gut pathogens, including *Streptococcus* spp. and *Clostridium* spp. Additionally, a microbial cluster originating as early as the acute infection phase was identified and characterized by elevated abundances of *R. gnavus*, *Escherichia coli*, *Veillonella* spp., and *Streptococcus* spp., along with decreased levels of *F. prausnitzii* and *Eubacterium* spp. [29]. This dysbiotic profile was significantly associated with reduced lung function at six months post-infection, including lower forced expiratory volume (FEV1), forced vital capacity (FVC), and impaired carbon dioxide diffusion. Notably, individuals in this microbial cluster also experienced more severe acute illness, with higher rates of intensive care unit admission and longer hospital stays during acute infection. Similarly, in another study, *R. gnavus* and *Clostridium* spp. were associated with a higher incidence of cardiopulmonary symptoms, including shortness of breath, cough, nasal congestion, and chest pain. Collectively, the literature consistently highlights the persistence of proinflammatory microbial genera, particularly *Ruminococcus*, *Streptococcus*, and *Clostridium*, as potential contributors to the development and maintenance of long COVID symptoms.

Patients experiencing gastrointestinal symptoms during long COVID have also demonstrated distinct alterations in gut microbiota composition [46]. Notably, genera such as *Neisseria*, *Lautropia*, and *Agrobacterium* were elevated, along with associated toxic serum metabolites, including 4-chlorophenylacetic acid, 5-sulfoxymethylfurfural, and estradiol valerate, which may contribute to gastrointestinal disturbances. Another study investigated microbial associations with specific symptom clusters and provided further insight into those related to gastrointestinal and fatigue-related symptoms [10]. The gastrointestinal symptom cluster was associated with significant increases in bacterial families such as Lachnospiraceae, Ruminococcaceae, Erysipelotrichaceae, Erysipelatoclostridiaceae, and Anaerovoracaceae. In contrast, the fatigue-only symptom group showed associations with *Lactobacillus*, *Limosilactobacillus*, *Sellimonas*, *Staphylococcus*, Acutalibacteraceae, and Anaerovoracaceae. No significant microbial associations were observed for cardiopulmonary, musculoskeletal, or neuropsychiatric symptom clusters in this study. Conversely, certain bacterial genera, including *Alistipes*, *Barnesiella*, and *Gordonibacter*, have shown negative correlations with gastrointestinal symptoms, suggesting potential protective roles [47]. Despite these emerging findings, current studies remain inconsistent in identifying specific microbial taxa consistently associated with gastrointestinal symptoms in long COVID, highlighting the need for further targeted investigation.

### 2.6. Methodological and Clinical Heterogeneity in COVID-19 Microbiome Studies

Inconsistent findings have been reported in the microbiome studies of COVID-19, and one of the driving forces is the vast difference in approaches and clinical variables across studies. One should also recognize that these differences may result in inconsistent outcomes and make any direct comparisons or meta-analyses difficult [48]. Various microbiome profiling techniques have been applied in different studies, yielding varying taxonomic resolution and organism types identified depending on the technique; typically, 16S rRNA gene amplicon sequencing offers a low cost, but low-resolution, taxonomic (and functional) resolution, typically just distinguishing bacteria at the genus level [49]. By comparison, shotgun metagenomic sequencing provides a much wider range of microbes, including viruses and fungi, and can determine bacteria to the species/strain level with metabolic gene content, though at a higher cost [50]. Due to these variations, 16S vs. shotgun studies might not reveal the same organisms or pathways to be important. As an example, a species-scale change observed through deep shotgun sequencing may just be called a genus-scale pattern in a study based on 16S [51]. These methodological inconsistencies may result in inconsistent conclusions regarding what microbial taxa are altered in COVID-19.

The composition of the gut microbiome changes throughout the infection and recovery period of COVID-19, thus the time of sample collection is highly relevant. Patient-centered studies (e.g., in acute COVID-19, i.e., with admission or within the first few weeks of the disease) frequently describe a severe dysbiosis, including decreased diversity and disappearance of beneficial taxa, at the acute end of infection [52]. It is noteworthy that in one meta-analysis, the alterations of the gut microbiome are the greatest approximately 7–30 days after the diagnosis (acute phase) when the microbiome starts to restore the more favorable state [53]. During the convalescent phase (weeks to months after overcoming the virus), much of the acute changes can be ameliorated as the microbiome resumes its functionality. In fact, patients who have completely healed COVID-19 (with no long COVID manifestations) are more likely to restore their gut flora to a more normal state a few months later, as opposed to patients with long COVID who may have dysbiosis months later [54]. Thus, a study that samples patients in active infection, will inherently have a different microbial profile than one that samples individuals in later recovery. Acute vs. convalescent timepoints represent different states of the microbes, and the lack of considering this can lead to the contradictory nature of the results between studies.

Antibiotics are frequently administered to COVID-19 patients, notably those admitted to hospitals, and this is a severe confounding variable. Antibiotics are commonly known to alter the gut microbiome, leading to massive declines in diversity and community composition changes that persist as long as weeks or months [55]. Actually, most patients with COVID-19 admitted to hospitals are estimated to have received empirically broad-spectrum antibiotics, even in case of low confirmed rates of bacterial co-infection. This may remove commensal bacteria to permit the proliferation of opportunistic organisms, such as Enterococcus or other hospital-associated organisms, to prevail [56]. Various studies differed in this aspect, with some eliminating patients who had recently taken antibiotics, others including them and this can result in strikingly different microbiome results. As an example, a cohort with a large proportion of patients on antibiotics may experience a greater loss in gut flora, due to drugs rather than the virus per se, whereas a cohort of patients with mild COVID-19 with no antibiotics may have less dramatic changes. This variability in the control of antibiotic application among studies therefore leads to disparate findings. It has been observed that one common exclusion criterion in microbiome studies is antibiotic use within 3 months of sampling since the gut community typically requires approximately 1.5 months to recover after antibiotics [54,57]. Unless this is factored in, the impact of COVID-19 on the microbiome is confounded by the perturbations caused by medication. Microbiome outcomes greatly depend on the clinical severity of the patients under study. A large number of initial COVID-19 microbiome studies targeted hospitalized patients with severe disease, which tends to be severely dysbiotic (loss of diversity and beneficial microbes, growth of proinflammatory taxa) and correlates with their extreme inflammatory condition. Conversely, other studies of mild or asymptomatic COVID-19 patients have found significantly smaller changes in microbiota, and in some studies, no statistically significant difference in the overall community structure between cases and healthy controls. As an example, a study found that even mild/asymptomatic COVID-19 patients were found to have an altered gut community composition compared to controls, although other studies found little or no global alteration in mild cases [11,58,59]. These differences suggest that the microbiome effect of SARS-CoV-2 may rely on the severity of illness. When one paper analyses patients in ICU and the other mostly minor outpatients, their conclusions could be at odds. Clinical heterogeneity in the form of only severe vs. mostly mild cases confined to various cohorts complicates cross-study analysis. It highlights the importance of tracking severity as a variable severe illness tends to be accompanied by factors such as intensive care, antibiotics, and high inflammation, which induce larger microbiome changes [54].

The other variable as the pandemic went on was whether the subjects were vaccinated against COVID-19 before sampling. Vaccination can tune the gut microbiome at least temporarily. Recently, it was demonstrated that the post-vaccination change in diversity and composition of gut microbes was observed in people after COVID-19 vaccination several weeks afterward [60]. This can be an expression of short-term immune activation or other post-vaccine physiological alterations. which can also account for microbial changes A report highlighted that pandemic-period study populations were an un-homogenic group of unvaccinated, vaccinated, infected previously, and never-infected persons and that it was no longer easy to determine which microbiome changes were attributable to COVID-19 and not to pre-existing immunity or other factors [61]. This may have resulted in misleading results between studies since it failed to stratify or control analysis based on the vaccination status to date. Thus, the history of vaccination and infection of participants should be included in the analysis to minimize this source of variation [54,62].

Diet, genetics, and environmental exposures contribute to the baseline gut microbiota variability in various populations and regions. Studies of the microbiome in relation to COVID-19 have been performed in various world regions (Asia, Europe, North America, etc.), whose normal microbiomes have their own profiles and diets that are predominant. Therefore, mechanisms of specific taxa change that are evident in one country might not entirely be replicated in another. In addition, the pandemic led to lifestyle changes in different parts of the world (lockdowns, changes in diet, hygiene), which could affect the outcomes. According to a 2022 meta-analysis, the broad difference in the intestinal microbiota of populations, diets, and lifestyles resulted in the inability to interpret individual research results in a coherent manner due to the wide variety of research approaches and design [48,63]. Such heterogeneity implies that a study may claim an increase in a particular genus of bacteria in COVID-19 patients, whereas a different study in a different location may also not detect the growth or even may reverse the trend of that same genus, merely due to differences in the population structures and conditions underlying such studies [64]. Ideally, meta-analyses or cross-cohort studies must consider such differences at the population level (e.g., by focusing on each cohort separately or adding diet/location as a covariate). Thus these various types of heterogeneity may be responsible for the discrepancies in the results of COVID-19 microbiome studies published and makes it impossible to meaningfully compare on a one-to-one basis, and makes meta-analysis more difficult, as combining data across studies without considering heterogeneity may mask real signals [48,65]. In light of these shortcomings, more recent work has attempted to reconcile and re-examine data on several cohorts using standardized techniques in order to discover any consistent microbial signature linked to COVID-19 that can be observed in various environments [66]. In the future, it will be vital that authors comment on these variables and where feasible, control or stratify them during study design and analyses. In this way, the discipline can shift towards more reproducible and generalizable conclusions regarding the role of the gut microbiome in COVID-19.

## 3. Proposed Mechanisms Linking Microbiome to Long COVID

There are several proposed mechanisms through which gut dysbiosis, or other microbiomes may be associated with development and persistence of long COVID symptoms.

### 3.1. Gut–Immune Axis and Inflammation

COVID-19 infection has been shown to exert long-lasting effects on the inflammatory profile of patients, altering both innate and adaptive immune responses [67]. The gut and immune system are closely interconnected, with the gut microbiota playing a central regulatory role. In patients with long COVID, a deficiency in butyrate-producing microbes has been observed alongside reduced levels of butyrate and other anti-inflammatory metabolites [2]. This imbalance, combined with an overrepresentation of proinflammatory Gram-negative bacteria, may sustain a state of chronic low-grade inflammation [6,45]. Notably, elevated levels of *Prevotella* and *Veillonella* in the oral and gut microbiomes have been associated with the activation of Toll-like receptor pathways, leading to increased production of proinflammatory cytokines such as IL-6, IL-1β, and IL-23. Additionally, dysbiosis may compromise intestinal barrier integrity, resulting in microbial translocation, commonly referred to as “leaky gut”, where microbial components (e.g., LPS, fungal antigens) enter the bloodstream [40,43,68,69,70]. Elevated levels of microbial translocation markers have been detected in long COVID patients and are associated with heightened immune activation [44,71,72,73,74,75]. Together, these immune-microbiome interactions may help explain the persistence of symptoms in long COVID, even after the virus itself has been cleared from the body (Figure 2).

Recent studies have linked certain gut microbes with markers of inflammation in post-acute COVID-19 patients that develop neuropsychiatric symptoms [76]. Specifically, decreased abundance of the *Akkermansia* genus was demonstrated to be correlated with post-acute COVID-19 patients, with concurrent increases in myeloid progenitor inhibitory factor 1 (MPIF-1) and interleukin (IL)-17 [76]. MPIF-1 is a chemokine that plays a role in immune regulation and can be elevated in response to proinflammatory stimuli from microbial derivatives such as LPS [76,77], while IL-17 is a proinflammatory cytokine that promotes neutrophil recruitment [78]. *Akkermansia*, supports mucosal barrier integrity, preventing serum translocation of inflammatory metabolites and exerting strong anti-inflammatory effects within the gut ecosystem [79]. Further studies have better characterized the cytokine and chemokine profiles in patients with long COVID that had associated proinflammatory shifts in gut microbial composition [80]. Over time, it was noted that a majority of chemokines such as MCP-1, MIP-1a and TNF-α may normalize by one year, while others such as GFF-2, G-CSF and IL-15 remained elevated, indicating dynamic changes in immune regulation over time. For example, *Eubacterium halii*, demonstrated a significantly negative correlation with the circulating levels of MCP-1, MiP-1b, IFN-α2, IL-10, IL-8, among others, suggesting a potential protective association of this microbial species [80]. IL-6, which has been known to serve as a central mediator of the acute-phase response in COVID-19, remained elevated throughout the one-year study period. Interestingly, this has been noted in another study of long COVID patients with predominant fatigue and gastrointestinal symptoms [81]. These patients also displayed persistent IL-6 elevation with gastrointestinal barrier dysfunction measured by markers of barrier leakage such as LPS response via lipopolysaccharide binding proteins (LBP). LBP are acute phase reactants that bind to LPS, the outer membrane of Gram-negative bacteria, which is associated with increased markers of barrier permeability, metabolic endotoxemia and systemic inflammation [82]. IL-6 is an end-product of the immune activation of these pathways, with increased systemic IL-6 response consistently shown to be associated with the development of long COVID [83].

**Figure 2 ijms-26-10120-f002:**
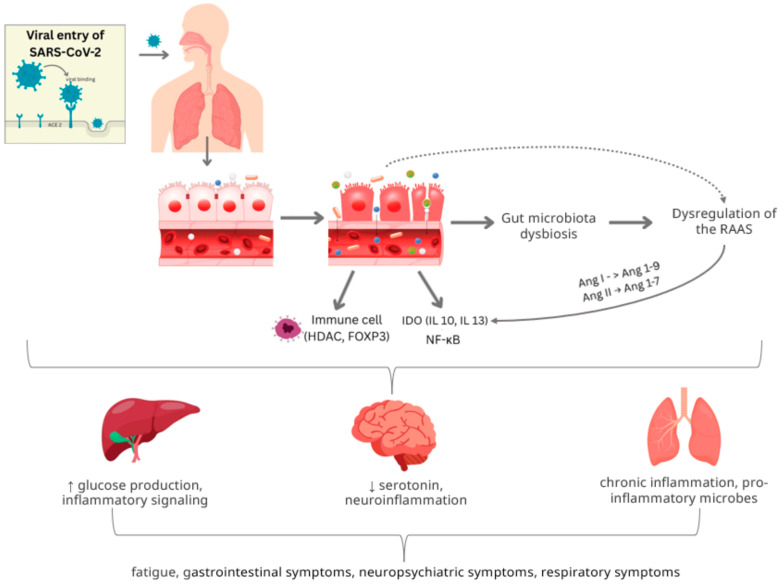
Interaction between SARS-CoV-2 infection, intestinal dysbiosis, and chronic inflammation in long COVID. SARS-CoV-2 enters host cells through ACE2 receptors in the respiratory and intestinal epithelium, leading to intestinal barrier disruption. Microbial translocation and altered gut signaling activate immune cells (HDAC, FOXP3) and induce IDO expression with downstream cytokines IL-10 and IL-13. In the liver, viral and immune-mediated effects promote glucose production and amplify inflammatory signaling. Dysregulation of the renin–angiotensin–aldosterone system (RAAS), including imbalances in Ang I → Ang 1–9 and Ang II → Ang 1–7 conversion, triggers NF-κB activation and sustains proinflammatory pathways. Continuous arrow represents a pathway primarily supported by human studies, while dotted arrow represents a pathway primarily supported by animal models. Abbreviations: ACE2, angiotensin-converting enzyme 2; HDAC, histone deacetylase; FOXP3, forkhead box P3; IDO, indoleamine 2,3-dioxygenase; IL-10, interleukin-10; IL-13, interleukin-13; RAAS, renin–angiotensin–aldosterone system; Ang I, angiotensin I; Ang II, angiotensin II; Ang 1–9, angiotensin 1–9; Ang 1–7, angiotensin 1–7; NF-κB, nuclear factor kappa-light-chain-enhancer of activated B cells; ↑: increase; ↓: decrease [44,84].

### 3.2. ACE2 and Gut Barrier Dysfunction

Angiotensin-converting enzyme 2 (ACE2), which is abundantly expressed in the intestinal epithelium, serves as a key entry receptor for SARS-CoV-2. Beyond facilitating viral entry, ACE2 plays a crucial role in regulating gut microbiota composition and maintaining intestinal barrier integrity through its influence on amino acid transport and antimicrobial peptide production [84]. In severe COVID-19, downregulation of gut ACE2 may contribute to microbial dysbiosis and impaired barrier function [85]. Interestingly, murine studies have shown that intestinal overexpression of ACE2 is associated with increased abundance of anti-inflammatory microbial species, in contrast to ACE2 knockout models, which exhibit more proinflammatory profiles [86]. Additionally, *Bacteroides* spp. have been shown to downregulate ACE2 expression in the murine gut, and their abundance inversely correlates with fecal SARS-CoV-2 viral load [9]. One proposed mechanism is that reduced ACE2 expression leads to altered secretion of antimicrobial peptides, contributing to the loss of beneficial microbes and disruption of microbial balance [21]. Once the gut barrier is compromised, translocation of microbial products and inflammatory mediators may drive progression to more severe and persistent disease states [87]. Moreover, SARS-CoV-2 infection has been linked to changes in gut microbiota, and these dysbiotic changes are associated with ongoing inflammation and tissue damage, even after viral clearance [88,89].

### 3.3. Microbiota-Gut–Brain Axis

Many symptoms of long COVID are neurological (e.g., brain fog, cognitive impairment) or neuropsychiatric (e.g., anxiety, insomnia). Certain gut microbes are known to communicate with the central nervous system via the gut–brain axis, using neural, hormonal, and immune pathways [90]. Increasing evidence suggests that post-COVID-19 dysbiosis may contribute to neurological and autonomic dysfunctions associated with long COVID [91]. One proposed mechanism involves disruption of the hypothalamic–pituitary–adrenal (HPA) axis due to altered gut microbiota and reduced vagal nerve stimulation, as observed in long COVID patients [21]. In practical terms, an imbalanced gut microbiome may impair the body’s ability to regulate stress responses and immune function, potentially resulting in anxiety, sleep disturbances, and cognitive deficits [92,93,94]. Supporting this, studies have shown that targeting the microbiome, particularly with *Bifidobacterium*, may improve cognitive function in individuals with mild cognitive impairment [95,96,97,98]. Additionally, SARS-CoV-2-induced intestinal barrier disruption via ACE2 interaction, along with systemic cytokine responses, may facilitate the entry of neurotoxic metabolites into the bloodstream [99]. These circulating toxins, originating from the gut, could damage cerebral vasculature and contribute to the persistence of neuropsychiatric symptoms in long COVID.

Microbial metabolites also play a significant role in regulating neurotransmitter levels, particularly in long COVID, where alterations in microbial composition can disrupt precursor availability and biosynthetic pathways [100]. For instance, reductions in *Bifidobacterium* and *Faecalibacterium*, both commonly depleted in long COVID, have been shown to affect tryptophan metabolism [21]. Tryptophan is a critical amino acid that undergoes hydroxylation to form 5-hydroxytryptophan (5-HTP), which is subsequently decarboxylated to produce serotonin (5-HT) [100]. Notably, over 90% of the body’s serotonin is synthesized in the gut. Recent studies have demonstrated that gut microbiome-derived L-tryptophan biosynthesis is reduced two-fold in individuals with prominent gastrointestinal symptoms during the acute phase and in those experiencing persistent mental health symptoms in the post-acute phase [101]. Among patients with long-term neuropsychiatric symptoms, serotonin levels were five times lower than levels in those with gastrointestinal symptoms alone. Serum levels of tryptophan and indole-3-propionate, a microbial metabolite of tryptophan, negatively correlate with proinflammatory cytokines during both the acute and post-acute phases of COVID-19 [102]. Fecal microbiota analysis in these patients also showed a depletion of microbial enzymes involved in tryptophan synthesis and an overall reduction in tryptophan metabolism compared to healthy controls. Collectively, these findings suggest that microbiota-mediated disruptions in tryptophan metabolism are closely associated with the persistence of neuropsychiatric symptoms in long COVID.

### 3.4. Viral Persistence, Immune Priming and Autoantibody Formation

Evidence suggests that the body’s microbiome may influence both viral clearance and the recovery process following SARS-CoV-2 infection. In some individuals, viral RNA has been detected in stool samples for months after the initial infection, and a subset of long COVID patients continue to exhibit detectable viral antigens in the gut mucosa [9]. The persistence of these antigens can sustain adaptive immune activation, including the stimulation of mucosal-associated invariant T (MAIT) cells, which may contribute to ongoing symptoms characteristic of long COVID [103]. The gut microbiota may further influence this process by cross-reacting with MAIT cells, promoting their activation and perpetuating inflammatory responses [104]. The development of autoantibodies is increasingly recognized as a potential contributor to the pathogenesis of long COVID [105]. Autoantibody production has been correlated with a range of post-acute symptoms, including respiratory, vasomotor, and neurological manifestations [105]. Persistent viral antigens and the presence of SARS-CoV-2 superantigens may promote autoantibody formation through the induction of polyclonal T-cell and B-cell activation [106,107].

Mechanisms such as molecular mimicry and epitope spreading are thought to underlie this immune dysregulation. In particular, several gut microbial species may contribute to immune priming or autoimmunity, potentially driving long COVID symptoms. Molecular mimicry occurs when bacterial proteins or metabolites share structural similarities with host or viral antigens, leading to inappropriate immune activation and loss of self-tolerance [108]. For example, neurological symptoms in post-acute COVID-19 have been associated with increased CD4+ T cell responses and reduced CD8+ T cell activity targeting the SARS-CoV-2 nucleocapsid protein [109], highlighting an imbalance in immune regulation that may be influenced by both viral and microbial factors. Gut microbiota play a critical role in modulating T-cell function, and disruptions in microbial composition can impair immune tolerance. A reduction in tolerogenic bacterial species may increase the risk of autoimmunity by weakening regulatory T cell (Treg) responses [110,111]. Some researchers propose that molecular mimicry where microbial antigens share structural similarities with host proteins can mislead the immune system into targeting self-antigens, thereby promoting autoimmune responses in long COVID [112]. Microbial dysbiosis is associated with immune dysregulation through the alteration of key metabolites, including SCFAs, secondary bile acids, tryptophan, and indole derivatives, all of which are crucial for maintaining Treg function and mucosal immune tolerance [113]. In parallel, as mentioned earlier, dysbiosis may modulate ACE2 expression and compromise gut epithelial integrity. As previously discussed, the overgrowth of inflammatory bacteria disrupts intestinal barrier function, facilitating systemic inflammation and autoimmune activation. Collectively, these disturbances can lead to sustained activation of both innate and adaptive immune responses, elevated autoantibody production, and enhanced antigen presentation of host proteins misidentified as foreign, thereby contributing to the pathogenesis of long COVID.

### 3.5. The Oral–Lung Aspiration Axis

Changes in the oral microbiome following COVID-19 infection may contribute to persistent respiratory symptoms in the post-acute phase. Bacterial aspiration has been observed in COVID-19 patients and is thought to promote chronic pulmonary inflammation during recovery [40,108]. Individuals with elevated levels of proinflammatory oral microbiota, particularly from the *Prevotella* and *Veillonella* genera, have been shown to be significantly more likely to develop prolonged symptoms and progress to long COVID [33]. These genera are known producers of endotoxins such as LPS, which stimulate the release of proinflammatory cytokines. Notably, *Veillonella* has been associated with elevated interleukin-6 (IL-6) levels, a cytokine commonly increased in both acute and long COVID cases [114,115]. Colonization of the respiratory tract by such “inflammatory-type” microbial communities may hinder respiratory recovery and potentially contribute to long-term complications such as pulmonary fibrosis [116]. This may help explain the frequently observed association between chronic respiratory issues and dysbiosis in both the gut and oral microbiomes [117]. Furthermore, the oral microbiota profiles of long COVID patients closely resemble those observed in individuals with chronic fatigue syndrome, suggesting a possible microbial link underlying the overlapping symptoms between these two conditions [33]. Additionally, SARS-CoV-2 has been shown to infect and replicate within the oral epithelium, with oral dysbiosis potentially associated with enhanced viral entry through upregulation of ACE2 expression mediated by bacterial enzymes that mimic furin activity [118]. These proteases produced by oral pathogens such as *Porphyromonas gingivalis* can activate the SARS-CoV-2 spike protein, enhancing infectivity [119]. Moreover, oral dysbiosis creates microenvironments that impair mucosal immunity, support viral persistence, and may contribute to the establishment of oral viral reservoirs [118]. Persistent SARS-CoV-2 infection and related symptoms such as dysgeusia, xerostomia, and periodontitis have been found to parallel these microbial alterations [120]. Collectively, these findings suggest that a balanced oral microbiome may play a protective role against the development of long COVID by supporting mucosal immune defense, reducing chronic inflammation, and limiting the overgrowth of proinflammatory taxa such as *Veillonella* and *Prevotella*. Conversely, oral dysbiosis appears to contribute to viral persistence and the manifestation of long COVID sequelae.

### 3.6. Experimental Evidence: Causal Role of Microbiota

Experimental studies have provided preliminary evidence suggesting a potential causal involvement of the microbiota in long COVID. In one study, germ-free mice that received fecal transplants from long COVID patients developed symptoms resembling those observed in human long COVID, suggesting that microbial dysbiosis may contribute directly to disease pathogenesis [121]. Notably, administration of *Bifidobacterium longum* to these mice partially reversed cognitive impairments and respiratory dysfunction, indicating the potential therapeutic effects of specific beneficial microbes [2,121]. These findings support the hypothesis that an imbalanced microbiota can drive symptoms associated with long COVID, although further research is needed to fully elucidate the underlying mechanisms. In summary, alterations in the microbiome may influence long COVID through several mechanisms, including sustained systemic inflammation, dysregulated immune responses, impaired mucosal barrier function, and disrupted communication along the gut–organ axes (e.g., gut–brain, gut–lung) [122]. A compromised intestinal barrier, or “leaky gut,” may allow microbial products to enter systemic circulation, triggering neuroinflammation and contributing to multi-organ dysfunction [123]. While the exact pathways remain under investigation, consistent evidence suggests that a diverse and balanced gut microbiota supports recovery, whereas microbial disruptions may prolong illness and delay resolution of symptoms following SARS-CoV-2 infection [21,33,124].

The proposed mechanisms connecting the microbiome to long COVID, as well as associated findings and limitations, are detailed in Table 2.

## 4. Microbial Biomarkers of Long COVID

As distinctions in the microbiome between individuals with long COVID and those who fully recover become more apparent, researchers are beginning to identify potential microbial biomarkers, such as specific taxa, microbial signatures, or metabolites, that may help predict the risk of developing long COVID or serve as diagnostic indicators. Several candidate biomarkers have already been proposed.

### 4.1. Reduced SCFA Producers

As previously discussed, reduced levels of butyrate-producing gut bacteria, such as *Faecalibacterium*, *Eubacterium*, *Subdoligranulum*, *Anaerostipes*, and *Bifidobacterium* species, have been commonly observed in individuals with long COVID patients [13,28]. In clinical practice, the detection of low abundances of these beneficial anaerobes in stool samples, particularly when accompanied by reduced overall microbial diversity, may signal an increased risk of developing long COVID [29]. One review highlighted a strong correlation between diminished levels of butyrate-producing bacteria and the presence of long COVID symptoms at six months post-infection [17]. As such, a “butyrate-producer deficit” may serve as a valuable microbial signature for identifying or predicting long COVID.

### 4.2. Enriched Pathobionts

An overabundance of proinflammatory microbes may serve as a negative indicator. Commonly observed taxa in individuals with long COVID include *R. gnavus* and *B. vulgatus* in the gut [2,17], and *Prevotella* and *Veillonella* in the oral cavity [33]. These species are frequently associated with an “inflammation-associated” microbiota profile [125]. Haran et al. specifically referred to the oral microbiome of long COVID patients as exhibiting “inflammation-type dysbiosis” [33]. From a clinical perspective, stool or saliva samples dominated by these proinflammatory taxa, particularly in the context of persistent symptoms, may suggest ongoing post-COVID-19 pathology [30,80]. For instance, the *Prevotella*-to-*Bifidobacterium* ratio in saliva and the *R. gnavus* to *F. prausnitzii* ratio in stool have been proposed as potential microbial biomarkers, pending further validation of their diagnostic relevance [16].

### 4.3. Microbiome Diversity Index

Alpha diversity, which reflects the richness and evenness of the microbial community, has emerged as a potential biomarker of gut microbial health and risk for progression to long COVID [126]. Individuals with long COVID often exhibit significantly lower gut microbiome diversity compared to healthy controls or those who have fully recovered [11,66]. In a one-year follow-up study by Zhang et al., alpha diversity metrics were able to differentiate individuals with long COVID from those without [13]. Routine analysis of 16S rRNA gene sequencing could be incorporated into clinical panels to assess whether a patient’s microbiome is returning to a healthy state following SARS-CoV-2 infection, potentially aiding in early identification and management of long COVID.

### 4.4. Specific Species or Functions

In addition to taxonomic shifts at the genus level, researchers have identified specific bacterial species and functional gene groups as potential indicators of long COVID. For example, a study from Hong Kong found that *C. innocuum* and *A. naeslundii* were frequently associated with fatigue and neurological symptoms in hospitalized patients [17]. *R. gnavus* has also been highlighted for its proinflammatory role, as it produces bacterial polysaccharides linked to inflammatory bowel disease and is commonly found in high abundance in stool samples from individuals with chronic COVID-19 syndrome [17,127,128]. Metagenomic analyses suggest that the long COVID microbiome may be enriched in genes involved in LPS production and other harmful metabolites, while showing a depletion of genes responsible for synthesizing SCFAs [129]. In the SIM01 trial, individuals who showed clinical improvement exhibited fewer gut microbial genes related to antibiotic resistance and a greater abundance of genes involved in the production of beneficial metabolites [32]. These functional changes may reflect the restoration of a healthier gut microbial community during recovery.

### 4.5. Combined Microbiome Signatures

A microbial signature that integrates the characteristics of multiple microorganisms is currently being developed using machine learning approaches. For example, Haran et al. employed random forest analysis to classify patients based on the duration of their COVID-19 symptoms, which may help in distinguishing individuals with different post-infection trajectories [33]. The integration of data from various omics platforms, such as metagenomics, metabolomics, and transcriptomics, may further enhance the development of robust biomarker panels for early prediction of long COVID [130]. Although this approach is still in its early stages, the goal is to create diagnostic tools, such as stool-based tests administered at the onset of COVID-19 that could help identify microbial patterns associated with long COVID risk and enable timely supportive interventions [17,131]. A recent News Medical report highlighted the potential of microbiome profiling to identify high-risk individuals even at the time of hospital admission [132]. It is important to note, however, that these microbiome-based biomarkers are not yet in clinical use. Additionally, certain microbes proposed as markers, such as low levels of *Faecalibacterium*, are also observed in other chronic conditions, which may limit specificity. Nonetheless, these findings are generating valuable insights. Future research will be critical in identifying more specific and clinically applicable microbiome-based markers for long COVID diagnosis and management [133]. At present, there is broad consensus that long COVID is associated with a decrease in beneficial commensal bacteria and an increase in proinflammatory microbial species [21,134]. This dysbiotic profile has been associated with long COVID and may represent a potential focus for future diagnostic research (Table 3).

## 5. Microbiome-Targeted Diagnostics and Therapeutics

Given the growing body of evidence suggesting associations between gut microbiota imbalances and long COVID, researchers and clinicians are increasingly investigating how the microbiome can be leveraged both as a diagnostic tool and as a target for therapeutic intervention.

### 5.1. Diagnostic Avenues

Emerging research suggests that analyzing a patient’s microbiome may aid in identifying cases of long COVID and individuals at elevated risk. For example, a stool-based diagnostic test could be developed to assess the abundance of key microbial taxa, akin to a “Long COVID Dysbiosis Index.” A reduced presence of butyrate-producing bacteria, coupled with a high abundance of pathobionts in stool samples from post-COVID-19 patients, may indicate the presence of long COVID and associated symptoms [135]. In cases with persistent respiratory or systemic complaints, elevated levels of oral pathobionts such as *Prevotella* and *Veillonella* may also serve as informative biomarkers. Although no clinically validated microbiome-based diagnostic is currently available, pilot studies suggest that such an approach is feasible. Specific microbiome signatures have been shown to correlate with the likelihood of developing long-term complications after SARS-CoV-2 infection [17]. Integrating microbiome sequencing data with other biomarkers, such as reactivation of Epstein–Barr virus or the presence of autoantibodies, may offer a more comprehensive risk assessment and help guide personalized treatment strategies.

### 5.2. Prebiotics and Diet

High-fiber, anti-inflammatory diets and fermented foods may help restore gut microbiota balance, with dietary interventions and prebiotics showing promising effects in patients with long COVID [136,137,138]. Among these, the Mediterranean diet, rich in fruits, vegetables, whole grains, nuts, seeds, olive oil, and herbs, has been associated with improved metabolic health in individuals recovering from COVID-19 [139]. Recent studies have proposed mechanisms through which the Mediterranean diet may reduce inflammation in long COVID patients, particularly through its antioxidant properties [140]. For example, redox interactions involving olive oil and lactate dehydrogenase were found to be mediated by gut *Oscillobacter* species. Interestingly, individuals with higher adherence to the Mediterranean diet had lower levels of *Oscillobacter* and more favorable lactate dehydrogenase concentrations [140]. Although the available data remain limited, prebiotics, the non-digestible fibers that serve as substrates for beneficial gut bacteria, have shown encouraging results in the context of COVID-19 [141]. Diets rich in fermentable fibers and omega-3 fatty acids may support the growth of SCFA-producing microbes, thereby reducing inflammation [21]. Such fibers are found in foods like inulin, oats, garlic, onions, and other oligosaccharide-rich compounds [141]. In murine models, supplementation with the prebiotic inulin improved COVID-19 outcomes by modifying gut microbial composition and increasing levels of deoxycholic acid [142].

Furthermore, the addition of polyphenols and other phytochemicals from fruits and vegetables to probiotic regimens resulted in a two-fold reduction in fatigue, three-fold reduction in cough, and two-fold improvement in quality-of-life scores, compared to probiotics alone [143]. Plant-based polyphenols and other compounds have attracted interest in in vitro and computational antiviral activity against SARS-CoV-2. These agents used in conjunction were found to have greater effects in sedentary, older, previously hospitalized men with gastrointestinal symptoms. Interestingly, phytochemicals such as curcumin are shown to mitigate oxidative stress through controlling elevations of reactive oxygen species (ROS) [143]. This was demonstrated via introduction of curcumin into COVID-19 infected cells, which increased NRF2 gene expression and restored NQO1 activity. NRF2 is a master regulator of the antioxidant response, with activation of antioxidant enzymes such as NQO1, which contributes to cellular defense by detoxifying ROS producing species [144]. Moreover, curcumin in particular has been researched, with pre-, co-, and post-infection treatment with curcumin (~10 µg/mL) in Vero E6 cell assays demonstrating approximately 99% viral inhibition with both D614G and Delta variants, and suppressed proinflammatory cytokines (IL-1, IL-6, IL-8) in PBMC assays [145]. In addition, a curcumin-based oro-nasal film spray prevented SARS-CoV-2 at low micromolar concentrations (EC 5 = 3.15 µg/mL) without causing epithelial cell death, and it also induced the expression of antimicrobial peptides and markers of innate mucosal immunity [146]. Curcuminoids (e.g., Me23) in neuronal models revealed an antiviral effect on SARS-CoV-2 replication, inhibition of oxidative stress by upregulating NRF2, and viral entry proteases (TMPRSS2/11D) [147]. Curcumin has been proposed to act through several antiviral and immunomodulatory mechanisms: spike or nucleocapsid binding, viral proteases (e.g., 3CL).

Beyond curcumin, there have been other phytochemicals including terpenoids and flavonoids such as luteolin, quercetin, kaempferol and wogonin that have shown roles in attenuating severity of COVID-19 and its sequelae [148]. A flavonol, kaempferol, which is abundant in a variety of fruits and vegetables, also demonstrates promising anti-SARS-CoV-2 activity. In vitro and in silico Kaempferol blocked spike-mediated membrane fusion by disrupting the heptad repeat fusion machinery and viral proteases such as the main protease (M 7). Other antiviral effects of kaempferol, not limited to SARS-CoV-2, such as inhibition of oxidative stress and inflammatory signals (e.g., NF-kB, MAPK), may potentially reduce tissue damage in response to viral infection [149]. Pinosylvin is a stilbenoid structurally related to resveratrol and has had less research in relation to SARS-CoV-2. It has shown antimicrobial and anti-inflammatory actions in other environments, and has been identified in phytochemical screening studies as a prospective binding to viral proteins [150]. Nevertheless, COVID-19 has limited direct in vitro or animal data and thus remains speculative and indicative of future studies. It should be highlighted that the vast majority of information regarding these phytochemicals is based on in vitro system, molecular docking, or mechanistic modeling. Their bioavailability, metabolic stability, tissue distribution, particularly to respiratory mucosa, and safe and effective dosing in humans continue to be significant challenges. In the case of curcumin, a number of approaches, such as nano-formulations, encapsulation, and conjugation, are under consideration to enhance its delivery in vivo [66]. In short, curcumin and kaempferol are the most actively proven phytochemicals with promising mechanisms to disrupt SARS-CoV-2 infection and regulate downstream inflammation. Pinosylvin is an interesting candidate but is not well validated as an antiviral agent. Although these phytochemicals show promise, their roles in the post-acute phase are not fully elucidated. While further research is needed to fully understand the therapeutic role of diet, phytochemicals and prebiotics in long COVID, the use of synbiotics, combinations of prebiotics and probiotics, has gained stronger support in the literature and will be addressed in the following subsection.

### 5.3. Probiotics and Synbiotics

Probiotics and synbiotics have emerged as promising therapeutic strategies for a range of health conditions, including long COVID. Accumulating evidence suggests that these interventions help restore gut microbial balance and promote a favorable intestinal microenvironment similar to that observed in healthy individuals [136,151]. One notable example is the SIM01 study conducted in Hong Kong, where a six-month supplementation with a *Bifidobacterium*-based synbiotic was associated with improvements in fatigue, mental fog, and gastrointestinal symptoms in long COVID patients compared to placebo [32]. The intervention enriched the gut microbiome, particularly increasing the abundance of SCFA-producing bacteria. Importantly, specific probiotic strains were associated with targeted benefits, for instance, increased levels of *B. longum* were associated with improved cognitive function and concentration. Similarly, another study reported that three months of supplementation with a synbiotic formulation containing *Lactobacillus rhamnosus*, *Lactobacillus plantarum*, *Bifidobacterium lactis*, *B. longum*, fructooligosaccharides, and zinc resulted in a significant reduction in fatigue during the post-acute phase of COVID-19 [152]. These improvements were accompanied by decreased post-exertional malaise and increased concentrations of neurotransmitters in brain regions such as the thalamus and frontal lobes. Interestingly, administering probiotics during the acute phase of COVID-19 may also reduce the risk of developing chronic fatigue symptoms later on [153]. In one study, hospitalized patients who received probiotics showed significant increases in beneficial metabolites such as arginine, asparagine, and lactate, along with reductions in harmful compounds like 3-hydroxyisobutyrate, a metabolite associated with obesity, inflammation, and insulin resistance [154]. At follow-up, a significantly lower proportion of patients in the probiotic group reported fatigue symptoms. Moreover, probiotics have been proposed as modulators of the gut-lung axis in long COVID. Following six months of probiotic supplementation, increases in stool concentrations of dopamine metabolites and butyrate were observed, alongside partial improvements in quality of life. Taxonomic shifts in key microbial families such as *Ruminococcaceae* and *Lachnospiraceae*, as well as beneficial modulation of immune responses, were noted in the intervention group. Taken together, current evidence supports the role of probiotics and synbiotics in alleviating multiple symptoms associated with long COVID, including fatigue, cognitive dysfunction, gastrointestinal distress, and systemic inflammation.

While early clinical findings regarding probiotics, synbiotics, dietary modulation, and microbiota transplantation in long COVID are promising, it is important to emphasize that these interventions remain preliminary. Current studies are limited by heterogeneous trial designs, small sample sizes, and short follow-up durations. Moreover, many rely on subjective symptom scales rather than objective physiological or biochemical endpoints, which may overestimate treatment effects. More long-term, multicenter randomized controlled trials are required to confirm both efficacy and safety, as well as to determine optimal formulations, dosing strategies, and treatment durations. Future investigations should harmonize outcome measures, longer monitoring periods, and integration of objective biomarkers (e.g., inflammatory or metabolic indices). Until such evidence is available, microbiome-targeted interventions for long COVID should be considered experimental and supportive rather than definitive therapeutic options. Clear communication of these uncertainties is critical to avoid overstating clinical effectiveness and to maintain scientific rigor in emerging microbiome research.

### 5.4. Microbiota Transplantation

Microbiota transplantation (MT) is a therapeutic approach designed to restore gut microbial balance by transferring microbiota from a healthy donor to a recipient [155]. While MT is currently approved primarily for the treatment of *Clostridioides difficile* infection, it is also under investigation for a variety of conditions associated with gut dysbiosis, including long COVID [156,157,158]. For instance, recent clinical trials evaluating encapsulated microbiota transplant in patients with post-COVID-19 symptoms such as fatigue and insomnia reported significant benefits compared to controls [31]. After 12 weeks, the insomnia severity index improved markedly in the FMT group, with 38% achieving remission versus 10% in the control group. These improvements were accompanied by reductions in blood cortisol levels, and recipients’ gut microbiota profiles shifted to more closely resemble those of healthy donors. Another study involving patients with mild to moderate COVID-19 symptoms demonstrated that MT also led to improvements in gastrointestinal symptoms, including diarrhea, as well as neuropsychiatric symptoms such as depression [159]. While these preliminary findings are promising, further well-controlled clinical trials are required to establish the safety, efficacy, and long-term outcomes of MT in the context of long COVID. Should MT prove effective and safe, it could potentially serve as a last-resort intervention to regenerate the microbiome in patients with persistent and debilitating symptoms. However, several challenges remain, including donor selection, regulatory approval, and patient acceptance of the procedure. At present, the use of FMT for long COVID remains experimental, as no formal regulatory guidelines have been established to support its clinical use in this context [160].

An overview of recent studies on microbiome-targeted therapies, their main outcomes, and limitations is presented in Table 4.

## 6. Consensus and Controversies

As research into the microbiome’s role in long COVID continues to advance, a growing scientific consensus is beginning to emerge, though several areas of uncertainty and debate remain.

### 6.1. Consensus Points

There is widespread agreement that gut microbiome disruption is present in a substantial proportion of individuals with long COVID. Multiple independent studies across diverse geographic regions, including China, Hong Kong, the United Kingdom, and the United States, have consistently shown that patients with persistent post-COVID-19 symptoms exhibit reduced abundance of beneficial microbial species and increased levels of opportunistic organisms in their gut microbiota [13,15,17,161]. For instance, a study conducted in Wuhan reported that recovered patients with ongoing symptoms had significantly lower microbial diversity and reduced levels of short-chain fatty acid-producing species compared to healthy controls [13]. Similarly, a U.S. study found that long COVID patients had higher abundances of *Bacteroides* and *Blautia*, alongside decreased levels of *Bifidobacterium* and *Faecalibacterium* [162]. In Hong Kong, researchers observed that specific symptom clusters were associated with particular microbial taxa: respiratory symptoms correlated with *R. gnavus* and *B. vulgatus*, while neuropsychiatric symptoms and fatigue were linked to *C. innocuum* and *A. naeslundii*. Notably, butyrate-producing bacteria were inversely correlated with symptom severity, underscoring their potential protective role [17]. These consistent findings across diverse populations suggest that, in individuals with long COVID, the gut microbiome often fails to fully recover to its pre-infection state. This persistent dysbiosis may contribute to ongoing symptoms and delayed recovery. Moreover, the extent of microbial imbalance has been shown to correlate with symptom severity, reinforcing the hypothesis that the gut microbiota plays an active and potentially causal role in the clinical trajectory of long COVID.

Growing evidence supports several mechanisms through which gut dysbiosis may contribute to the development and persistence of long COVID. These include sustained low-grade systemic inflammation, dysregulated immune responses, and disruptions in gut-derived metabolic and signaling pathways, particularly those involving the gut–brain and gut–lung axes [75,93]. In light of these proposed mechanisms, microbiome-targeted interventions have been explored. Preliminary studies using probiotics and microbiota transplantation suggest that restoring beneficial microbial populations can lead to partial improvements in clinical symptoms, further supporting the role of the gut microbiome in modulating disease expression. Most of these studies have consistently utilized specific probiotic strains such as *B*. *longum*, *L. rhamnosus*, *L. plantarum*, and *B. lactis*, which appear central to reestablishing microbial balance and alleviating symptoms. Common intervention periods range from 3 to 6 months, during which participants have reported notable improvements in fatigue, cognitive function, and gastrointestinal symptoms. This consistency in both probiotic strain selection and treatment duration suggests a potentially reproducible therapeutic approach for managing long COVID. Moreover, these beneficial effects have been linked to increased levels of SCFA-producing bacteria and favorable modulation of immune responses, reinforcing the hypothesis that microbiota restoration is a critical component of post-COVID-19 recovery [32,136,152,153,154]. In summary, the role of the microbiome in long COVID has gained broad scientific acceptance, marking a shift from early skepticism to the recognition of dysbiosis as a key contributor to the condition’s pathophysiology.

### 6.2. Controversies and Open Questions

Despite growing recognition of the microbiome’s involvement in long COVID, several key questions remain unresolved. A central point of debate concerns the directionality and causality of the relationship: Does dysbiosis contribute to the development of long COVID, or is it a consequence of lifestyle changes, reduced physical activity, altered diet, and pharmacological treatments during and after SARS-CoV-2 infection? For example, a study in the adult population of Saudi Arabia found that during COVID-19 quarantine, over 40% of participants reported increased food and snack consumption, leading to weight gain in 28% of respondents. Additionally, 52% reported a decline in physical activity [163]. Similarly, a study conducted in Egypt found that 60% of participants consumed fast food, and 52.7% reported low adherence to the Mediterranean diet. Physical activity declined in 61% of participants, accompanied by a significant increase in body mass index [164]. These behavioral and dietary changes are known to affect the gut microbiota, raising the possibility that observed dysbiosis may, at least in part, reflect these confounding factors.

Another layer of complexity involves pharmacological interventions. To date, no universally effective antiviral therapy has been established for SARS-CoV-2, due in large part to the virus’s rapid mutation rate. As a result, treatment remains largely symptomatic, relying on various medications that may inadvertently impact gut microbial balance [165]. For example, while the effects of the antiretroviral combination lopinavir/ritonavir on the gut microbiota of COVID-19 patients are not fully characterized, studies in HIV-1-infected individuals receiving this regimen have demonstrated significant alterations in microbial composition. These include shifts in the major bacterial phyla Firmicutes, Proteobacteria, Bacteroidetes, and Actinobacteria, along with marked reductions in beneficial genera such as *Lachnospira*, *Butyricicoccus*, *Oscillospira*, and *Prevotella* [166]. Taken together, these findings underscore the need for careful interpretation of microbiome studies in long COVID, with consideration of confounding lifestyle and treatment-related factors. Establishing causality will require longitudinal, controlled studies that account for these variables.

Most studies caution that most existing evidence on the microbiome and long COVID are observational in nature and, therefore, cannot establish causality. While experimental models, such as fecal microbiota transplantation in animals, offer intriguing insights, their applicability to humans remains debated. Moreover, findings across studies often differ in terms of which microbial taxa are implicated. Some studies report significant changes in microbial diversity, while others observe minimal shifts, particularly in smaller cohorts. These discrepancies may be attributed to differences in geographic location, dietary patterns, environmental exposures, or host genetics. This variability has sparked ongoing debate about whether dysbiosis follows a universal pattern in long COVID or manifests differently across populations. Additionally, it remains unclear whether dysbiosis is a persistent condition or one that gradually resolves over time. Some evidence suggests that microbiome imbalances in long COVID patients may normalize beyond one-year post-infection [21,167]. However, other studies indicate that dysbiosis can persist for 18 months or longer, with a subset of patients continuing to exhibit symptoms and microbial imbalance for years, similar to observations in individuals with chronic fatigue syndrome [13]. These uncertainties highlight the need for long-term, longitudinal studies to better understand the trajectory and clinical significance of post-COVID-19 dysbiosis.

Another area of ongoing debate concerns the extent to which microbiome imbalance contributes to long COVID symptoms relative to other factors. Many researchers consider alternative mechanisms, such as autoimmunity and viral persistence in tissue reservoirs, as potentially more significant drivers of long COVID pathology. While some investigators view microbiome alterations as a central factor, others regard them as secondary or even incidental. This divergence in perspective is reflected in treatment strategies: some clinicians prioritize microbiome-targeted therapies, while others use them adjunctively or not at all. There is also debate over whether observed differences in microbiome composition translate into clinically meaningful effects. For example, results from synbiotic intervention trials suggest potential benefits; however, most improvements were reported through self-assessment measures rather than objective clinical endpoints. Critics of such studies have noted that the reported gains in daily functioning were modest and difficult to interpret due to the subjective nature of symptom reporting. As a result, many healthcare professionals remain cautious, calling for larger, higher-powered studies with standardized outcome measures before endorsing microbiome-based treatments for long COVID [168].

Some studies argue that methodological differences have significantly shaped the debate around the role of the microbiome in long COVID. At the time many early studies were conducted, a standardized definition of long COVID had not yet been established, leading researchers to include heterogeneous patient populations in their analyses [6,169,170,171]. Although consensus definitions have since emerged, typically defining long COVID as symptoms persisting beyond 2–3 months, variations still exist across studies. Additionally, the use of different microbiome profiling techniques, such as 16S rRNA sequencing and shotgun metagenomics, can yield divergent results, further complicating comparisons. The need for high-quality, standardized studies across multiple research centers is frequently emphasized, yet remains challenging to achieve. There is also ongoing debate about how the microbiome interacts with other health conditions. For example, questions persist about whether microbiome imbalances can trigger autoimmune diseases or, conversely, whether autoimmune conditions can alter the microbiome. Similarly, among the many symptoms of long COVID, it remains unclear whether cardiovascular manifestations are more closely linked to microbiome changes than neurological ones, questions that continue to be actively investigated [26]. While there is broad agreement in the scientific community that microbiome alterations are common in long COVID and likely contribute to its pathophysiology, uncertainty remains regarding the strength, consistency, and clinical implications of these changes. A 2025 review emphasized the need for future research to evaluate the long-term consequences of gut dysbiosis and to determine whether interventions such as dietary modification or probiotic supplementation can effectively mitigate these effects [21]. This reflects both the growing optimism around targeted microbiota therapies and the many unresolved questions that still surround this emerging field.

## 7. Conclusions

The relationship between the microbiome and long COVID has rapidly evolved from a speculative hypothesis to a dynamic and growing area of scientific inquiry. Over the past few years, accumulating evidence has shown that alterations in gut and, to a lesser extent, oral microbiota often persist in individuals experiencing post-COVID-19 symptoms. These changes, particularly the depletion of beneficial short-chain fatty acid-producing bacteria and the enrichment of proinflammatory taxa, provide a plausible framework for understanding the chronic inflammation and multi-systemic manifestations characteristic of long COVID. Disruptions in gut barrier function, imbalances in microbial metabolites, and aberrant immune activation have all been proposed as mechanisms linking gut dysbiosis to long COVID pathophysiology. Although it remains premature to assert a direct causal relationship, dysbiosis appears to be a significant contributing factor that interacts with viral persistence, immune dysregulation, and possibly psychosocial stressors. From a clinical perspective, the microbiome offers promising avenues for therapeutic intervention. Preliminary studies involving probiotics, synbiotics, and FMT suggest that partial restoration of microbial balance may alleviate certain symptoms of long COVID. These findings offer hope that adjunct therapies targeting the microbiome could enhance recovery and improve quality of life in affected patients. Looking ahead, more rigorous and large-scale clinical trials will be essential to validate these early findings and to define the most effective microbiome-modulating strategies. Furthermore, integration of microbiome profiling into long COVID risk assessment and management may allow for more personalized and preventative approaches. In summary, emerging evidence underscores the complex interplay between SARS-CoV-2 infection, host immune responses, and the gut microbiota in shaping outcomes in the post-acute phase. Restoring microbial balance may prove to be a key component of long COVID recovery, but continued research is needed to elucidate underlying mechanisms and optimize microbiota-based interventions.


## Figures and Tables

**Figure 1 ijms-26-10120-f001:**
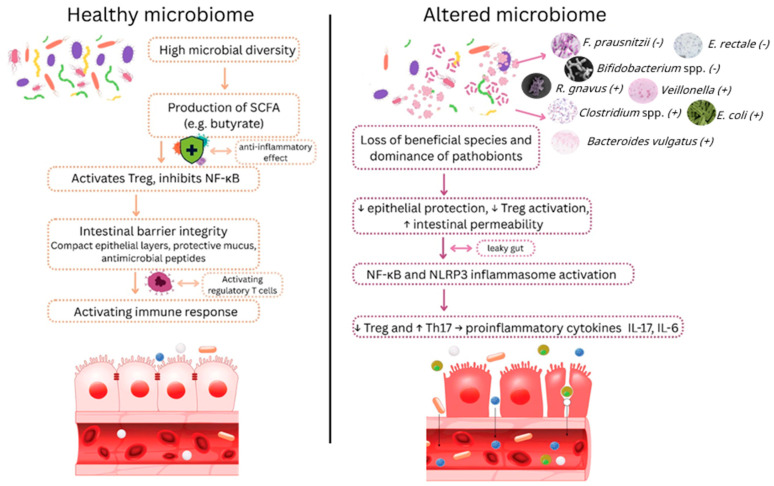
The contrasting microbial profiles observed in individuals who recover fully after SARS-CoV-2 infection versus those who develop long COVID. Patients who recover typically demonstrate restoration of a balanced gut microbiota, with microbial diversity and relative abundances resembling those of healthy, uninfected controls. In contrast, individuals with long COVID exhibit persistent dysbiosis characterized by reduced alpha diversity and depletion of beneficial, short-chain fatty acid–producing taxa such as *Faecalibacterium prausnitzii*, *Eubacterium rectale*, and *Bifidobacterium* spp. Concurrently, there is an enrichment of proinflammatory and opportunistic pathobionts, including *Ruminococcus gnavus*, *Bacteroides vulgatus*, *Veillonella*, and certain *Clostridium* species. These taxonomic shifts have been associated with chronic low-grade inflammation, impaired intestinal barrier integrity, and prolonged systemic symptoms. The visually specified taxa represent the species whose alteration has been most consistently reported in multiple clinical studies investigating dysbiosis associated with long COVID, particularly in the chronic period of 3–6 months post-infection. The schematic representation of the altered gut microbiome (**bottom right**) illustrates the loss of epithelial barrier integrity, evidenced by widened intercellular spaces (“leaky gut”), and increased inflammatory cell infiltration into the lamina propria, resulting from the activation of the NF-κB and NLRP3 pathways, emphasizing the inflammatory state and increased permeability. Abbreviations: SCFA, short-chain fatty acid; Treg, regulatory T cell; NF-κB, nuclear factor kappa-light-chain-enhancer of activated B cells; NLRP3, NOD-, LRR-, and pyrin domain-containing protein 3; Th17, T helper 17 cells; IL-17, interleukin-17; IL-6, interleukin-6; −: decrease; +: increase; ↑: increase; ↓: decrease [8,13,17].

**Table 1 ijms-26-10120-t001:** Summary of selected key studies (2019–2024) on the relationship between the microbiome and long COVID-19.

Year (Ref.)	Sample (Design)	N	Study Country	Time Since Infection (Months)	Sequencing Method	Microbiome Focus	Key Taxa Alterations	Clinical Outcomes	Main Findings	Limitation
2025 Blankestijn et al. [29]	Post-COVID-19 clinic cohort 3–6 months post-infection; fecal metagenomic sequencing with clustering analysis	79	Netherlands	3–6	Shotgun metagenomics	Gut	↓ *F. prausnitzii*, *Eubacterium* spp.; ↑ *R. gnavus*, *E. coli*, *Veillonella*, *Streptococcus*	Pulmonary function (FEV_1_, FVC, DLCO); severity correlation	Patients with dysbiotic microbiota had poorer lung recovery; dysbiosis linked to severe acute COVID-19 history.	Small single-country study; observational; confounders (diet, meds) not fully controlled.
2024 Su et al. [30]	Multi-cohort machine learning study predicting PACS phenotypes from microbiome data	Several hundred	China (Hong Kong)	3–6	Shotgun metagenomics	Gut	Enterotypes: one depleted in butyrate-producers, enriched in *R. gnavus*, *E. coli*	PACS symptom clusters (respiratory, neuro, GI)	Gut enterotypes predicted symptom type with ~89% accuracy; microbiome heterogeneity underpins symptom diversity.	Complex model; regional limitation; predictive, not causal.
2024 Lau et al. [31]	Non-randomized open-label trial: 30 received multi-donor FMT (capsules), 30 received control	60	China (Hong Kong)	9	Shotgun metagenomics	Gut	↑ *Gemmiger formicilis*, ↑ donor-like microbiota composition	Insomnia, fatigue, anxiety, cortisol levels	FMT improved sleep quality and reduced fatigue/anxiety; safe with no serious AEs.	Open-label, moderate N, short follow-up; limited generalizability.
2023 Lau et al. [32]	RCT: Long COVID patients randomized 1:1 to synbiotic (SIM01) vs. placebo for 6 months	463	China (Hong Kong)	4–10	Shotgun metagenomics	Gut	↑ *Bifidobacterium adolescentis*, SCFA-producers; ↑ diversity	Symptom improvement (fatigue, memory loss, GI issues, well-being)	Synbiotic improved symptoms and microbiome diversity; microbiome correlated with recovery.	Subjective outcomes; single-center; no objective function gains.
2023 Zhang et al. [13]	Recovered patients 1 year post-hospitalization (84 long COVID, 103 recovered, 32 controls)	219	China (Wuhan)	12	16S rRNA sequencing	Gut	↓ *Eubacterium hallii*, *Subdoligranulum*, *Ruminococcus*, *Agathobacter*; ↑ *Veillonella*	Presence of long COVID symptoms at 12 months	Long COVID associated with reduced diversity and SCFA-producing genera; dysbiosis persisted ≥1 year.	Cross-sectional; single region; no diet control; association only.
2022 Liu et al. [17]	Prospective cohort followed from diagnosis to 6 months; pre-pandemic controls included	174	China (Hong Kong)	6	Shotgun metagenomics	Gut	↓ *Faecalibacterium*, *Bifidobacterium*; ↑ *R. gnavus*, *B. vulgatus*	Persistence of ≥1 symptom at 6 months (fatigue, respiratory, GI, neuro)	Baseline microbiota predicted long COVID; persistent dysbiosis at 6 months in long COVID.	Single-region cohort; no causal inference; did not adjust for acute severity.
2021 Haran et al. [33]	Adult COVID-19 outpatients; tongue swabs collected during acute illness; followed until symptom resolution (~37% developed long COVID)	27	USA	0–3	Shotgun metagenomics	Oral	↑ *Prevotella*, *Veillonella*	Symptom duration; presence of long COVID	Oral dysbiosis linked to prolonged inflammation; microbiota resembled chronic fatigue syndrome.	Small sample; no uninfected controls; observational study.
2020 Zuo et al. [9]	15 COVID-19 in-patients with varying severity; 15 uninfected controls; 6 pneumonia controls; longitudinal stool sampling during hospitalization	36	China (Hong Kong)	0–1	Shotgun metagenomics	Gut	↑ *Coprobacillus*, *C. ramosum*, *C. hathewayi*; ↓ *F. prausnitzii*	Acute COVID-19 severity; microbiome persistence post-clearance	Gut dysbiosis persisted from admission to discharge; opportunistic pathogens correlated with severity.	Small hospitalized cohort; no follow-up in mild/asymptomatic cases.

FMT: fecal microbiota transplantation; SCFA: short-chain fatty acid; RCT: randomized controlled trial; GI: gastrointestinal; ↑: increase; ↓: decrease.

**Table 2 ijms-26-10120-t002:** Summary of selected key studies of pathophysiologic mechanisms, important findings and limitations within studies.

Year (Ref.)	Sample (Design)	Study Subjects	Pathophysiologic Mechanism	Main Findings	Limitations
2025 Barichello et al. [76]	Cross-Sectional Study	Humans	Inflammation	Significant increases in MPIF-1, IL-1 and triglycerides in long-COVID19 patients. β-diversity was reduced, including decreased abundance of *Akkermansia* spp. No differences were observed in α-diversity data	Could not identify species in 16S rRNA analysis Cognitive assessment 3–4 weeks post-COVID-19 may be suboptimal Findings may be specific to study population or methodology Akkermansia depletion not consistently reported in other studies Psychiatric symptom assessment influenced by multifactorial factors
2025 Rohrhofer et al. [81]	Prospective Observational Study	Humans	Intestinal Barrier Disruption	Significant associations were present between gastrointestinal and neuropsychiatric symptoms in long COVID. In the post-acute phase, patients showed higher LBP/sCD14, lower IL-33 and higher IL-6 levels, indicating a proinflammatory state and intestinal barrier disruption	Low sample size Self-reported data
2024 Mussabay et al. [80]	Prospective Cohort Study	Humans	Inflammation	Severe COVID19, complicated by pneumonia, increased presence of proinflammatory bacterial species. In their post-acute phase various cytokines and chemokines such as MDC, IL-1b, TNF-α, FGF-2, EGF, IL-1RA, IFN-α, IL-10, sCD40L, IL-8, IL-12p40 and MIP-1b displayed a proinflammatory profile	Relatively small sample size (n = 60) Included patients that had severe COVID19 disease only
2024 Song et al. [86]	Experimental Study	Murine	ACE2	ACE2 knockout mice has increase inflammatory bacterial genera including *Deferribacteres*, *Parasutterella*, *Catenibacterium*, *Anaerotruncus*, with concomitant decreases in SCFA-producing bacteria. Contrarily, ACE2-overexpression enhanced concentrations of SCFA-producing bacteria such as *Lactobacillus*, *Bifidobacterium*, *Alisipes*, etc.	Study was performed in murine models and not in humans
2024 Yao et al. [102]	Experimental (Shotgun Metagenomic) with a Meta-Analysis	Human	Neurotransmitters Inflammation	Serum levels of tryptophan and IPA were negatively correlated with inflammatory markers such as circulating cytokines, while C-Trp, ILA and IAA were positively correlated with proinflammatory markers in long COVID patients. Metagenomics of microbiota showed reduction in enzymes involved in tryptophan metabolism in hospitalized patients. Microbiota-derived tryptophan metabolites modified TH1 and TH17 associated cytokine responses and reduced innate cell proinflammatory responses to TLR3 and TLR4	Only hospitalized patients included Confounding factors such as diet, medications, comorbidities and lifestyle were not adjusted for
2023 Visvabharathy et al. [109]	Observational, Cross-Sectional Study	Humans	Adaptive Immune Response	Patients with neurological symptoms in the post-acute COVID19 phase have elevated CD4 T cell response and reduced CD8 activation. CD8 T cell production of IL-6 heightened severity of neurologic symptoms	Small sample size Unable to control for time of sample collection with respect to date of COVID19 symptom onset
2022 Blackett et al. [101]	Randomized Control Trial	Human	Neurotransmitter Regulation	Gut microbiome L-tryptophan synthesis was decreased in patients with more severe gastrointestinal symptoms in acute COVID19. Similar biosynthesis pathways of tryptophan biosynthesis were also decreased in those with severe mental health symptoms in the post-acute phase.	Low sample size 5-HT concentrations were obtained postprandially 2 different cohorts were designed for separate studies, with differences in methods for self-reported mental health symptoms
2021 Haran et al. [33]	Prospective Cohort Study	Humans	Oral Microbiota Inflammation	Patients with prolonged COVID19 symptoms and progression to long COVID had higher abundance of proinflammatory microbiota including *Prevotella* and *Veillonella*. The oral microbiome in long COVID patients were similar to those with chronic fatigue syndrome	Sample size
2020 Gammazza et al. [107]	Bioinformatics Study	Computational	Molecular Mimicry	HSPs, which are considered human molecular chaperones, participate in molecular mimicry following COVID19 infection. Post-translational modifications can cause autoimmune endothelial damage in the post-acute phase	Hypothesis generated study Scanned for exact peptides No tissue confirmation

Abbreviations: MPIF-1: myeloid progenitor inhibitory factor 1; IL: interleukin; MDC: macrophage-derived chemokine; TNF: tumor necrosis factor; FGF: fibroblast growth factor; EGF: epidermal growth factor; IFN: interferon; CD40: cluster of differentiation 40; MIP-1b: macrophage inflammatory protein-1 beta; LBP: liposaccharide binding protein; ACE2: angiotensin converting enzyme 2; SCFA: short chain fatty acid; 5-HT: 5-hydroxytryptamine; IPA: indole-3-propionate; C-Trp: C-glycosyltrytophan; ILA: indole-3-lactic acid; IAA: indole-3-acetic acid; TLR: toll-like receptor; TH: t-helper cell; HSP: heat-shock proteins.

**Table 3 ijms-26-10120-t003:** Proposed microbial biomarkers of long COVID and their diagnostic relevance.

Category/Biomarker Type	Representative Studies (Year, Ref.)	Key Microbial Findings/Taxa or Functions	Potential Clinical or Diagnostic Relevance	Mechanistic or Functional Basis	Limitations/Validation Needs
Reduced SCFA Producers	Zhang et al., 2023 [13]; Liu et al., 2022 [17]; Zuo et al., 2020 [9]; Yeoh et al., 2021 [8]	↓ *Faecalibacterium*, *Eubacterium*, *Subdoligranulum*, *Anaerostipes*, *Bifidobacterium* spp.; reduced fecal butyrate levels and SCFA synthesis genes.	Low SCFA-producer abundance may signal increased risk of long COVID or delayed recovery; stool SCFA quantification could aid monitoring.	Loss of butyrate-producing bacteria → reduced anti-inflammatory and mucosal healing capacity → chronic inflammation and barrier dysfunction.	Observed in multiple cohorts but not specific to COVID; further validation in prospective studies needed.
Enriched Pathobionts	Haran et al., 2021 [33]; Liu et al., 2022 [17]; Su et al., 2024 [30]	↑ *Ruminococcus gnavus*, *Bacteroides vulgatus* (gut); ↑ *Prevotella*, *Veillonella* (oral); enrichment of proinflammatory Gram-negative species.	High abundance of these taxa correlates with persistent fatigue, GI, and neurological symptoms; may indicate ongoing mucosal inflammation.	Pathobiont overgrowth → LPS release → immune activation via TLR pathways; cytokine production (IL-6, IL-1β, TNF-α).	Requires cross-validation; ratios like *R. gnavus*, *F. prausnitzii* may improve specificity.
Microbiome Diversity Index	Zhang et al., 2023 [13]; Liu et al., 2022 [17]; Yeoh et al., 2021 [8]; Haran et al., 2021 [33]	Reduced alpha diversity (Chao1, Shannon, Simpson indices) in long COVID vs. recovered or control groups.	Alpha diversity metrics may serve as global indicators of gut health and recovery trajectory after COVID-19.	Lower diversity reflects ecosystem instability and loss of protective taxa; correlates with immune dysregulation and persistent symptoms.	Requires standardization; diversity thresholds vary by population and sequencing platform.
Specific Species or Functional Genes	Liu et al., 2022 [17]; Lau et al., 2023 [32]; Su et al., 2024 [30]	↑ *Clostridium innocuum*, *Actinomyces naeslundii*, *R. gnavus*; ↓ SCFA-related genes; ↑ genes for LPS and antibiotic resistance pathways.	Specific species and gene profiles may stratify long COVID subtypes (fatigue-dominant vs. neurocognitive).	Functional shift toward proinflammatory and oxidative pathways; depletion of metabolic and barrier-supportive functions.	Functional gene panels are promising but require longitudinal validation and standard bioinformatics pipelines.
Combined Microbiome Signatures	Haran et al., 2021 [33]; Su et al., 2024 [30]; Liu et al., 2022 [17]	Integrated microbial signatures combining taxa (e.g., *Prevotella*: *Bifidobacterium* ratio) and metabolic pathways via machine learning classifiers.	Predictive models could identify high-risk patients early; basis for stool- or saliva-based diagnostic assays.	Multi-omics integration (metagenomics + metabolomics + transcriptomics) enhances specificity for long COVID risk prediction.	Still experimental; reproducibility across populations and sequencing platforms remains unproven.

Abbreviations: SCFA: short-chain fatty acid; LPS: lipopolysaccharide; TLR: Toll-like receptor; GI: gastrointestinal; Chao1/Shannon/Simpson indices: alpha diversity metrics; TLR: Toll-like receptor; ↑: increase; ↓: decrease.

**Table 4 ijms-26-10120-t004:** Summary of selected key studies (2022–2025) of microbiome targeted therapies, important findings and limitations within studies.

Year (Ref.)	Sample (Design)	N	Microbiome-Targeted Therapy	Main Findings	Limitations
2025 Suarez-Morena et al. [139]	Cross-Sectional Study	305	Mediterranean Diet	Introduction of a Mediterranean diet in patients with Long COVID improved waist circumference and improved metabolic parameters	Sample size was predominately women and may not be generalizable to men Adherence was gathered through questionnaires
2024 Cuevas-Sierra et al. [140]	Prospective Cohort Study	188	Mediterranean Diet	High adherence to the Mediterranean diet improved inflammatory and oxidative markers. Adherence was also correlated with reduction in LDL-cholesterol and glucose levels. These changes were accompanied by lower *Oscillobacter* concentrations in high adherence groups which was related to oxidative markers.	Relatively small sample size Only some components of the Mediterranean Diet were used (i.e., olive oil)
2024 Song et al. [142]	Experimental	19	Prebiotics (Inulin)	Inulin supplementation improved concentrations of fecal SCFAs and secondary bile acids in the setting of COVID19	Experiment was performed in murine models Small sample size
2024 Lau et al. [32]	Double-Blind Placebo-Controlled Trial	463	Synbiotics (SIM01)	Six-month supplementation with a *Bifidobacterium*-based synbiotic significantly improved fatigue, mental fog, and gastrointestinal symptoms in long COVID patients compared to placebo	Difficult to assess based on severity of initial COVID-19 infection
2024 Ranisavljev et al. [152]	Double-Blind Placebo-Controlled Trial	26	Synbiotic	Three-month supplementation of synbiotics and placebo improved fatigue in Long COVID. Synbiotics attenuated post-exercise malaise. Synbiotics increased choline levels within the thalamus	Sample limited to young-to-middle-aged adults Vaccination status, time since infection, and dietary factors not considered Multi-strain synbiotic and placebo may have confounded effects Small sample size; short 3-month follow-up Limited mechanistic biomarker assessment
2024 Lau et al. [31]	Pilot Study	30	Fecal Microbiota Transplant	After 12 weeks, following FMT, the treatment group exhibited decreased symptoms associated with insomnia and anxiety. Serum cortisol levels were also lower. Depletion of microbial species that produce harmful derivatives were observed and microbiota was similar to donors at the study endpoint.	Low sample size
2024 Jiang et al. [159]	Double-Blind Placebo-Controlled Trial	40	Fecal Microbiota Transplant	FMT showed improvements in diarrhea, depression and neuropsychiatric symptoms in COVID-19. Serum AST/ALT ratio was reduced following FMT	Low sample size
2022 Santinelli et al. [153]	Retrospective Observational Study	58	Probiotics	Probiotic supplementation significantly lowered proportion of COVID-19 patients that reported fatigue. Concentrations of serum amino acids were increased and harmful microbiota byproducts were reduced	Lack of randomization
2022 Thomas et al. [143]	Double-Blind Placebo-Controlled Trial	147	Phytochemical and Probiotics	Addition of phytochemical two-fold reduction in fatigue, three-fold reduction in cough, and two-fold improvement in quality-of-life scores, compared to probiotics alone	Non-randomization of the probiotic element within the study

Abbreviations: LDL: low density lipoprotein; SCFA: short chain fatty acid; FMT: fecal microbiota transplantation; AST: aspartate aminotransferase; ALT: alanine aminotransferase.

## Data Availability

No new data were created or analyzed in this study. Data sharing is not applicable to this article.

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
