# Peer review of "Microbiome and Long COVID-19: Current Evidence and Insights"

_ijms, 2025, doi:10.3390/ijms262010120_

Round 1
Reviewer 1 Report
Comments and Suggestions for Authors
This review provides a clear and timely overview of how the gut and oral microbiomes may influence the development and persistence of long COVID. It explains how COVID-19 disrupts the balance of microbes—reducing beneficial bacteria and increasing those linked to inflammation—and connects these shifts to symptoms such as fatigue, brain fog, gut problems, and respiratory issues. The manuscript is well-structured, linking microbiome changes to mechanisms like impaired gut barriers, chronic inflammation, and immune disruption, while also considering possible treatments including probiotics, synbiotics, and fecal transplants. Helpful visuals and tables make the findings easier to follow, and the broad scope, balanced discussion of limitations, and inclusion of recent references ensure the review is both informative and credible for clinicians and researchers.
Weaknesses and Suggestions
- No conclusion section – The review ends with references without tying together the key messages. A short conclusion would strengthen the impact and clarify what we know, what we don’t, and what research is most needed.
- Repetition in mechanisms – Some microbiome changes (like loss of Faecalibacterium prausnitzii) are repeated without adding new insights. Consider condensing these sections and adding a visual (such as a pathway diagram) to explain mechanisms more clearly.
- Figure 1 clarity – The lower right corner of Figure 1 suggests less (dying) gut epithelial cells with reduced regeneration and less interaction with microbes compared to the healthy gut in the lower left. This could confuse readers. Updating the figure with clearer labeling or a short description would reduce misinterpretation.
Author Response
This review provides a clear and timely overview of how the gut and oral microbiomes may influence the development and persistence of long COVID. It explains how COVID-19 disrupts the balance of microbes—reducing beneficial bacteria and increasing those linked to inflammation—and connects these shifts to symptoms such as fatigue, brain fog, gut problems, and respiratory issues. The manuscript is well-structured, linking microbiome changes to mechanisms like impaired gut barriers, chronic inflammation, and immune disruption, while also considering possible treatments including probiotics, synbiotics, and fecal transplants. Helpful visuals and tables make the findings easier to follow, and the broad scope, balanced discussion of limitations, and inclusion of recent references ensure the review is both informative and credible for clinicians and researchers.
Weaknesses and Suggestions
- No conclusion section – The review ends with references without tying together the key messages. A short conclusion would strengthen the impact and clarify what we know, what we don’t, and what research is most needed.
R: We thank the reviewer for their positive feedback. A dedicated conclusion section (Section 7: Conclusions) is included in the manuscript, summarizing the key findings, knowledge gaps, and future research directions on the microbiome and long COVID.
- Repetition in mechanisms – Some microbiome changes (like loss of Faecalibacterium prausnitzii) are repeated without adding new insights. Consider condensing these sections and adding a visual (such as a pathway diagram) to explain mechanisms more clearly.
R: Thank you for this comment. While Faecalibacterium prausnitzii and other key taxa are mentioned in multiple sections, these repetitions are intentional, as each context highlights distinct aspects: acute-phase alterations, gut–brain axis effects, microbial biomarkers, combined microbiome signatures, and consensus points across studies. The repeated mentions serve to link mechanisms, clinical relevance, and biomarker interpretation, ensuring that readers can follow the role of these taxa across different pathways and stages of long COVID. To further enhance clarity, we have added Table 2: Summary of Selected Key Studies of Pathophysiologic Mechanisms, Important Findings, and Limitations, which provides a concise overview of each mechanism alongside study details and limitations, effectively complementing the textual descriptions.
- Figure 1 clarity – The lower right corner of Figure 1 suggests less (dying) gut epithelial cells with reduced regeneration and less interaction with microbes compared to the healthy gut in the lower left. This could confuse readers. Updating the figure with clearer labeling or a short description would reduce misinterpretation.
R: Thank you for this valuable comment. We have addressed the concern regarding Figure 1. The image in the bottom right corner, has been replaced with a clearer representation to avoid misinterpretation. In addition, we have complemented the figure legend with the following description to clarify the intended message: “The schematic representation of the altered gut (bottom right) illustrates the loss of epithelial barrier integrity, evidenced by widened intercellular spaces (‘leaky gut’), and increased inflammatory cell infiltration into the lamina propria, resulting from the activation of the NF-κB and NLRP3 pathways, emphasizing the inflammatory state and increased permeability.”
Reviewer 2 Report
Comments and Suggestions for Authors
The COVID-19 pandemic has highlighted not only the global impact of a viral pathogen but also the pivotal role of the human microbiome in influencing disease severity and recovery. This timely and well-crafted review summarizes the rapidly expanding literature on gut and oral microbiota in long COVID, discussing taxonomic alterations, proposed mechanisms such as SCFAs, tryptophan metabolism, barrier dysfunction, and immune priming, as well as potential therapeutic strategies including probiotics, synbiotics, and FMT. Importantly, the authors emphasize that causal relationships remain to be firmly established. Overall, the review provides clinicians and researchers with a clear and up-to-date conceptual overview.
Comments:
Microbiome findings in COVID-19 studies are often inconsistent due to methodological and clinical heterogeneity. Sequencing approaches (16S rRNA vs shotgun metagenomics) differ in taxonomic resolution, and sampling timepoints (acute vs convalescent) capture distinct microbial states. Additional variables such as antibiotic exposure, hospitalization, vaccination status, and geographic background further shape microbial composition. These differences contribute to conflicting results across studies and hinder meaningful meta-analysis, which could cause inconsistent conclusions reached by different studies and should be discussed in the text.
Figure 1 could be improved by labeling representative taxa on the schematic (e.g., Faecalibacterium increases; Ruminococcus gnavus decreases) and indicating sampling windows (acute vs 3–12 months post-infection). Including a small legend to denote evidence strength (e.g., taxa consistently vs inconsistently reported across studies) would further enhance clarity and informativeness.
Figure 2 would benefit from highlighting the strength of supporting evidence along each pathway. For example, clearly mark mechanisms supported by animal models or randomized clinical trials versus those with only limited mechanistic evidence from human studies. This distinction will help readers quickly gauge the robustness of each proposed link.
Table 1 could be expanded to include additional columns such as study country, time since infection (in months), sequencing method (16S vs shotgun), key taxa alterations, main clinical outcomes assessed, and the primary limitation of each study. These additions would facilitate clearer cross-study comparisons and help readers quickly identify sources of variability.
Author Response
The COVID-19 pandemic has highlighted not only the global impact of a viral pathogen but also the pivotal role of the human microbiome in influencing disease severity and recovery. This timely and well-crafted review summarizes the rapidly expanding literature on gut and oral microbiota in long COVID, discussing taxonomic alterations, proposed mechanisms such as SCFAs, tryptophan metabolism, barrier dysfunction, and immune priming, as well as potential therapeutic strategies including probiotics, synbiotics, and FMT. Importantly, the authors emphasize that causal relationships remain to be firmly established. Overall, the review provides clinicians and researchers with a clear and up-to-date conceptual overview.
Comments:
Microbiome findings in COVID-19 studies are often inconsistent due to methodological and clinical heterogeneity. Sequencing approaches (16S rRNA vs shotgun metagenomics) differ in taxonomic resolution, and sampling timepoints (acute vs convalescent) capture distinct microbial states. Additional variables such as antibiotic exposure, hospitalization, vaccination status, and geographic background further shape microbial composition. These differences contribute to conflicting results across studies and hinder meaningful meta-analysis, which could cause inconsistent conclusions reached by different studies and should be discussed in the text.
R: Thank you for this valuable suggestion. We have addressed this point by adding a new subsection entitled “2.6. Methodological and Clinical Heterogeneity in COVID-19 Microbiome Studies.” This section discusses the major sources of inconsistency across COVID-19 microbiome studies, including differences in sequencing approaches (16S rRNA vs. shotgun metagenomics), sampling timepoints (acute vs. convalescent), antibiotic exposure, disease severity, vaccination status, and geographic or population-specific factors. We also emphasize how these methodological and clinical variables can confound results and hinder meaningful meta-analysis, thereby aligning with the reviewer’s observation.
Figure 1 could be improved by labeling representative taxa on the schematic (e.g., Faecalibacterium increases; Ruminococcus gnavus decreases) and indicating sampling windows (acute vs 3–12 months post-infection). Including a small legend to denote evidence strength (e.g., taxa consistently vs inconsistently reported across studies) would further enhance clarity and informativeness.
R: The figure has been modified to indicate the taxa that increase or decrease using “+” or “–” symbols. Only taxa that have been consistently reported across multiple studies are included. To further clarify, we have updated the figure legend to state: “The visually specified taxa represent the species whose alteration has been most consistently reported in multiple clinical studies investigating the dysbiosis associated with long COVID, particularly in the chronic period of 3–6 months post-infection.” We did not include additional elements such as sampling timepoints directly in the figure, as these details are already presented in Table 1, which has also been updated with supplementary information, including post-infection periods and other relevant findings. These modifications improve the clarity and informational value of the figure while avoiding redundancy.
Figure 2 would benefit from highlighting the strength of supporting evidence along each pathway. For example, clearly mark mechanisms supported by animal models or randomized clinical trials versus those with only limited mechanistic evidence from human studies. This distinction will help readers quickly gauge the robustness of each proposed link.
R: Thank you for your suggestion. Figure 2 has been slightly revised following recommendations from another reviewer. In the updated version, the mechanisms are presented more clearly, illustrating both their interconnections and their contributions to the development of long COVID symptoms. In line with your recommendation, the level of evidence for each pathway is now highlighted. Additionally, the manuscript includes Table 2, which provides a comprehensive summary of key studies supporting the mechanisms shown in Figure 2. The table details study design, subjects, pathophysiologic mechanisms, main findings, and limitations, thereby reinforcing the evidence for the pathways illustrated in the figure.
Table 1 could be expanded to include additional columns such as study country, time since infection (in months), sequencing method (16S vs shotgun), key taxa alterations, main clinical outcomes assessed, and the primary limitation of each study. These additions would facilitate clearer cross-study comparisons and help readers quickly identify sources of variability.
R: Thank you for the helpful suggestion. We have expanded Table 1 to include the recommended columns, study country, time since infection, sequencing method, key taxa alterations, main clinical outcomes assessed, and primary study limitations, to enhance clarity and facilitate cross-study comparisons.
Reviewer 3 Report
Comments and Suggestions for Authors
I am attaching my comments in a separate Word file

Author Response
After going through the manuscript I believe it is very interesting and thorough, and very rigorous in its approach and bibliographical inclusion. I believe it needs only a few adjustments, namely:
- Some specific and technical terms like Chao index, Simpson index, etc., are not explained at first mention — I believe a brief definition, with the suitable references, at first mention, would greatly benefit the readers.
R: Thank you very much for your constructive comment and for your positive feedback on the manuscript. We have added the explanation for the Chao and the Simpson index. Short definitions, along with the corresponding references, have been added at their first mention in the text to improve clarity for the readers.
- The figures offer very detailed captions, which is a definite plus, but I would advise that the relative references for the mechanisms presented are included in the captions
R: We sincerely thank you for your valuable comment and for highlighting the detailed figure legends as a strength. We have revised the figures to include the relevant references directly in the legends, and additional improvements have also been made based on feedback from the other reviewers.
- For sections 3, 4, 5, and 6, comprehensive tables with the references (preferably classified thematically or by date) would be advisable to increase the presentability of the manuscript.
R: Thank you for this helpful suggestion. In response, we have added 3 tables to enhance clarity and presentability. Table 2 summarizes key studies of pathophysiologic mechanisms, including sample, study subjects, mechanisms, main findings, and limitations. Table 3 presents proposed microbial biomarkers of long COVID, their representative studies, key findings, clinical or diagnostic relevance, mechanistic basis, and validation needs. Table 4 summarizes microbiome-targeted therapies (2022–2025), including study sample, intervention, main findings, and limitations.
- It is interesting that you mention the role of foods and phytochemicals in preventing or mitigating the effects of COVID19. Could you perhaps expand this part a little more to specifically refer to phytochemicals which have a proven antiviral action against COVID19 like curcumin, pinosylvin and kaempferol
R: We have expanded Section 5.2 (Prebiotics and Diet) to provide a more detailed overview of plant-derived compounds and their antiviral and immunomodulatory properties in the context of COVID-19. The text now includes evidence on curcumin and kaempferol, which have demonstrated inhibitory effects on viral replication and modulation of inflammatory and oxidative pathways, as well as pinosylvin as a potential candidate. Limitations related to bioavailability, metabolic stability, and the preliminary nature of the data are highlighted to ensure a balanced and accurate presentation.
Reviewer 4 Report
Comments and Suggestions for Authors
At this stage, the review seems to capture the essence of a descriptive overview rather than a synthesis that is critical and original. It is recommended to revise it substantially before the work may be considered for publication. The major comments are the following:
The manuscript does not sufficiently highlight what sets it apart from previous reviews on the microbiome and long COVID. What new perspective does the review provide, including the integration of the gut–oral–lung axes, biomarker identification, and possibilities for translation?
In many cases, findings are presented without examining the study design, sample size, or other weaknesses. For instance, the review of Table 1 does not adequately address the small cohort size of several studies.
Numerous assertions suggest causation between discordant microbiomes and long COVID. However, this remains largely correlative. To accurately reflect the current state of evidence, it needs to revise these claims so that conclusions are not overstated.
The review discusses a number of possible mechanisms (the gut–immune axis, tryptophan metabolism, ACE2 regulation, autoimmunity), which are presented one after the other and are presented sequentially without integration. It is important to create an integrated model or diagram that illustrates the relationships between these mechanisms and their contributions to the pathophysiology of long COVID.
It is important to recognize the value in the discussion around probiotics, synbiotics, diet, and microbiota transplantation, but the projected effectiveness is presented with more certainty than current evidence supports. It is highly recommended to label these interventions as preliminary and to discuss their shortcomings (i.e. heterogeneous trial designs, subjective outcomes, lack of long-term data).
Although the reference list does include many relevant studies, more recent literature is missing, and the reference list has not been sufficiently updated (meaning important references from the last 2–3 years 2023–2025). While maintaining the relevance of the study citation, try to limit redundancy as some findings are surrounded by highly overlapping citations.
The manuscript is somewhat repetitive, though generally understandable. For instance, the depletion of Faecalibacterium prausnitzii is repeated several times with nearly identical wording. I encourage the author to consolidate these overlapping themes to enhance clarity.
A list of abbreviations would improve the readability of the paper due to the frequent use of acronyms (e.g. SCFA, LBP, ACE2, RAAS).
Summary
There is potential for this review to be of value, but it demands more compelling critical evaluation as well as clearer articulation of novelty, updated literature, and more balanced moderation of claims. I believe the manuscript, in its current form, has the potential to making an important contribution to the discipline.
Author Response
At this stage, the review seems to capture the essence of a descriptive overview rather than a synthesis that is critical and original. It is recommended to revise it substantially before the work may be considered for publication. The major comments are the following:
The manuscript does not sufficiently highlight what sets it apart from previous reviews on the microbiome and long COVID. What new perspective does the review provide, including the integration of the gut–oral–lung axes, biomarker identification, and possibilities for translation?
R: Thank you for this insightful observation. In response, we have reformulated the introduction to more clearly highlight what sets this review apart from previous work. Specifically, our review provides an integrative perspective on the microbiome in long COVID, explicitly considering the gut–oral–lung axes. It highlights novel candidate microbial biomarkers, including SCFA-producing bacteria, pathobionts, and functional gene shifts, with potential diagnostic and prognostic value. Additionally, we discuss translational applications, such as diet, pre/probiotics, synbiotics, and fecal microbiota transplantation, linking mechanistic insights to potential therapeutic strategies. This systems-level approach offers a more comprehensive framework for understanding host–microbiome interactions in long COVID and guiding future clinical interventions.
In many cases, findings are presented without examining the study design, sample size, or other weaknesses. For instance, the review of Table 1 does not adequately address the small cohort size of several studies.
R: Thank you for this observation. While Table 1 already includes a column summarizing the limitations reported by each study, for example, small sample sizes, single-region cohorts, observational or open-label designs, and lack of control for confounding factors such as diet or acute disease severity, to make this clearer we have added the following sentence before the Table 1: “The studies summarized in Table 1 provide evidence of persistent microbiome alterations in long COVID; however, many are limited by small sample sizes, observational designs, and lack of control for confounding factors, highlighting the need for cautious interpretation.”
Numerous assertions suggest causation between discordant microbiomes and long COVID. However, this remains largely correlative. To accurately reflect the current state of evidence, it needs to revise these claims so that conclusions are not overstated.
R: We agree that much of the current evidence linking microbiome alterations to long COVID is correlational rather than definitively causal. To address this, we have carefully revised the manuscript to replace statements implying direct causation with more precise language reflecting association, potential influence, or hypothesized contribution.
The review discusses a number of possible mechanisms (the gut–immune axis, tryptophan metabolism, ACE2 regulation, autoimmunity), which are presented one after the other and are presented sequentially without integration. It is important to create an integrated model or diagram that illustrates the relationships between these mechanisms and their contributions to the pathophysiology of long COVID.
R: We have revised Figure 2, which originally presented the mechanisms separately. In the updated version, the figure has been redesigned as an integrative diagram that not only illustrates each mechanism but also explicitly shows the connections and interactions between them. The new layout highlights the flow from SARS-CoV-2 entry through the intestine, the development of leaky gut and dysbiosis, subsequent immune activation and inflammatory signaling, and downstream effects on target organs (liver, brain, lungs) leading to the clinical manifestations of long COVID.
It is important to recognize the value in the discussion around probiotics, synbiotics, diet, and microbiota transplantation, but the projected effectiveness is presented with more certainty than current evidence supports. It is highly recommended to label these interventions as preliminary and to discuss their shortcomings (i.e. heterogeneous trial designs, subjective outcomes, lack of long-term data).
R: Thank you for highlighting this important point. We have revised Section 5.3 (Probiotics and Synbiotics) to clearly emphasize that these interventions remain preliminary. The updated text addresses the limitations of current studies. We also highlight the need for long-term, multicenter randomized controlled trials, harmonized outcome measures, and integration of objective biomarkers. Interventions are now presented as experimental and supportive, rather than definitive therapeutic options, ensuring accurate communication of current evidence and uncertainties.
Although the reference list does include many relevant studies, more recent literature is missing, and the reference list has not been sufficiently updated (meaning important references from the last 2–3 years 2023–2025). While maintaining the relevance of the study citation, try to limit redundancy as some findings are surrounded by highly overlapping citations.
R: We appreciate this observation. To address it, we have updated the reference list to include key studies from 2023–2025, ensuring that the review reflects the most recent evidence. We also streamlined citations to reduce redundancy, retaining only those that provide distinct or essential contributions.
The manuscript is somewhat repetitive, though generally understandable. For instance, the depletion of Faecalibacterium prausnitzii is repeated several times with nearly identical wording. I encourage the author to consolidate these overlapping themes to enhance clarity.
R: Thank you for this comment. While Faecalibacterium prausnitzii and other key taxa are mentioned in multiple sections, these repetitions are intentional, as each context highlights distinct aspects: acute-phase alterations, gut–brain axis effects, microbial biomarkers, combined microbiome signatures, and consensus points across studies. The repeated mentions serve to link mechanisms, clinical relevance, and biomarker interpretation, ensuring that readers can follow the role of these taxa across different pathways and stages of long COVID.
A list of abbreviations would improve the readability of the paper due to the frequent use of acronyms (e.g. SCFA, LBP, ACE2, RAAS).
R: A comprehensive list of abbreviations, covering all acronyms used throughout the manuscript, has been added to improve readability and facilitate understanding.
Round 2
Reviewer 3 Report
Comments and Suggestions for Authors
I sincerely thank the authors for implementing the requested changes in a timely and concise fashion. I wish them best of luck in their future endavours.